# Input Similarity from the Neural Network Perspective

Guillaume Charpiat[1]     Nicolas Girard[2]     Loris Felardos[1]     Yuliya Tarabalka[2,3]

[1] TAU team, INRIA Saclay, LRI, Univ. Paris-Sud
[2] TITANE team, INRIA Sophia-Antipolis, Univ. Côte d'Azur
[3] LuxCarta Technology
`firstname.lastname@inria.fr`

## Abstract

Given a trained neural network, we aim at understanding how similar it considers any two samples. For this, we express a proper definition of similarity from the neural network perspective (*i.e.* we quantify how undissociable two inputs $A$ and $B$ are), by taking a machine learning viewpoint: how much a parameter variation designed to change the output for $A$ would impact the output for $B$ as well?

We study the mathematical properties of this similarity measure, and show how to estimate sample density with it, in low complexity, enabling new types of statistical analysis for neural networks. We also propose to use it during training, to enforce that examples known to be similar should also be seen as similar by the network.

We then study the self-denoising phenomenon encountered in regression tasks when training neural networks on datasets with noisy labels. We exhibit a multimodal image registration task where almost perfect accuracy is reached, far beyond label noise variance. Such an impressive self-denoising phenomenon can be explained as a noise averaging effect over the labels of *similar* examples. We analyze data by retrieving samples perceived as similar by the network, and are able to quantify the denoising effect without requiring true labels.

## 1   Introduction

The notion of similarity between data points is an important topic in the machine learning literature, obviously in domains such as image retrieval, where images similar to a query have to be found; but not only. For instance when training auto-encoders, the quality of the reconstruction is usually quantified as the $L^2$ norm between the input and output images. Such a similarity measure is however questionable, as color comparison, performed pixel per pixel, is a poor estimate of human perception: the $L^2$ norm can vary a lot with transformations barely noticeable to the human eye such as small translations or rotations (for instance on textures), and does not carry semantic information, *i.e.* whether the same kind of objects are present in the image. Therefore, so-called *perceptual losses* [10] were introduced to quantify image similarity: each image is fed to a standard pre-trained network such as VGG, and the activations in a particular intermediate layer are used as descriptors of the image [4, 5]. The distance between two images is then set as the $L^2$ norm between these activations. Such a distance carries implicitly semantic information, as the VGG network was trained for image classification. However, the choice of the layer to consider is arbitrary. In the ideal case, one would wish to combine information from all layers, as some are more abstract and some more detail-specific. Then, how to choose the weights to combine the different layers? Would it be possible to build a canonical similarity measure, well posed theoretically?

More importantly, the previous literature does not consider the notion of input similarity from the point of view of the neural network that is being used, but from the point of view of another one (typically, VGG) which aims at imitating human perception. Yet, neural networks are black boxes difficult to interpret, and showing which samples a network considers as similar would help to explain its decisions. Also, the number of such similar examples would be a key element for confidence estimation at test time. Moreover, to explain the self-denoising phenomenon, *i.e.* why predictions can

be far more accurate than the label noise magnitude in the training set, thanks to a noise averaging effect over *similar* examples [11], one needs to quantify similarity according to the network.

The purpose of this article is to express the notion of similarity from the network's point of view. We first define it, and study it mathematically, in Section 2, in the one-dimensional output case for the sake of simplicity. Higher-dimensional outputs are dealt with in Section 3. We then compute, in Section 4, the number of neighbors (*i.e.*, of similar samples), and propose for this a very fast estimator. This brings new tools to analyze already-trained networks. As they are differentiable and fast to compute, they can be used during training as well, *e.g.*, to enforce that given examples should be perceived as similar by the network (*c.f.* supp. mat.). Finally, in Section 5, we apply the proposed tools to analyze a network trained with noisy labels for a remote sensing image alignment task, and formalize the self-denoising phenomenon, quantifying its effect, extending [11] to real datasets.

## 2 Similarity

In this section we define a proper, intrinsic notion of similarity as seen by the network, relying on how easily it can distinguish different inputs.

### 2.1 Similarity from the point of view of the parameterized family of functions

Let $f_\theta$ be a parameterized function, typically a neural network already trained for some task, and $\mathbf{x}$, $\mathbf{x}'$ possible inputs, for instance from the training or test set. For the sake of simplicity, let us suppose in a first step that $f_\theta$ is real valued. To express the similarity between $\mathbf{x}$ and $\mathbf{x}'$, as seen by the network, one could compare the output values $f_\theta(\mathbf{x})$ and $f_\theta(\mathbf{x}')$. This is however not very informative, and a same output might be obtained for different reasons.

Instead, we define similarity as the influence of $\mathbf{x}$ over $\mathbf{x}'$, by quantifying how much an additional training step for $\mathbf{x}$ would change the output for $\mathbf{x}'$ as well. If $\mathbf{x}$ and $\mathbf{x}'$ are very different from the point of view of the neural network, changing $f_\theta(\mathbf{x})$ will have little consequence on $f_\theta(\mathbf{x}')$. Vice versa, if they are very similar, changing $f_\theta(\mathbf{x})$ will greatly affect $f_\theta(\mathbf{x}')$ as well.

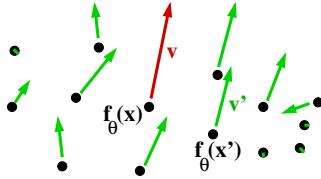

Figure 1: Moves in the space of outputs. We quantify the influence of a data point $\mathbf{x}$ over another one $\mathbf{x}'$ by how much the tuning of parameters $\theta$ to obtain a desired output change $\mathbf{v}$ for $f_\theta(\mathbf{x})$ will affect $f_\theta(\mathbf{x}')$ as well.

Formally, if one wants to change the value of $f_\theta(\mathbf{x})$ by a small quantity $\varepsilon$, one needs to update $\theta$ by $\delta\theta = \varepsilon \frac{\nabla_\theta f_\theta(\mathbf{x})}{\|\nabla_\theta f_\theta(\mathbf{x})\|^2}$. Indeed, after the parameter update, the new value at $\mathbf{x}$ will be:

$$f_{\theta+\delta\theta}(\mathbf{x}) \;=\; f_\theta(\mathbf{x}) + \nabla_\theta f_\theta(\mathbf{x}) \cdot \delta\theta + O(\|\delta\theta\|^2) \;=\; f_\theta(\mathbf{x}) + \varepsilon + O(\varepsilon^2).$$

This parameter change induces a value change at any other point $\mathbf{x}'$ :

$$f_{\theta+\delta\theta}(\mathbf{x}') \;=\; f_\theta(\mathbf{x}') + \nabla_\theta f_\theta(\mathbf{x}') \cdot \delta\theta + O(\|\delta\theta\|^2) \;=\; f_\theta(\mathbf{x}') + \varepsilon \frac{\nabla_\theta f_\theta(\mathbf{x}') \cdot \nabla_\theta f_\theta(\mathbf{x})}{\|\nabla_\theta f_\theta(\mathbf{x})\|^2} + O(\varepsilon^2).$$

Therefore the kernel $k_\theta^N(\mathbf{x}, \mathbf{x}') = \dfrac{\nabla_\theta f_\theta(\mathbf{x}) \cdot \nabla_\theta f_\theta(\mathbf{x}')}{\|\nabla_\theta f_\theta(\mathbf{x})\|^2}$ represents the influence of $\mathbf{x}$ over $\mathbf{x}'$: if one wishes to change the output value $f_\theta(\mathbf{x})$ by $\varepsilon$, then $f_\theta(\mathbf{x}')$ will change by $\varepsilon\, k_\theta^N(\mathbf{x}, \mathbf{x}')$. In particular, if $k_\theta^N(\mathbf{x}, \mathbf{x}')$ is high, then $\mathbf{x}$ and $\mathbf{x}'$ are not distinguishable from the point of view of the network, as any attempt to move $f_\theta(\mathbf{x})$ will move $f_\theta(\mathbf{x}')$ as well (see Fig. 1). We thus see $k_\theta^N(\mathbf{x}, \mathbf{x}')$ as a measure of similarity. Note however that $k_\theta^N(\mathbf{x}, \mathbf{x}')$ is not symmetric.

**Symmetric similarity: correlation** Two symmetric kernels natural arise: the inner product:

$$k_\theta^I(\mathbf{x}, \mathbf{x}') \;=\; \nabla_\theta f_\theta(\mathbf{x}) \cdot \nabla_\theta f_\theta(\mathbf{x}') \tag{1}$$

and its normalized version, the correlation:

$$k_\theta^C(\mathbf{x}, \mathbf{x}') = \frac{\nabla_\theta f_\theta(\mathbf{x})}{\|\nabla_\theta f_\theta(\mathbf{x})\|} \cdot \frac{\nabla_\theta f_\theta(\mathbf{x}')}{\|\nabla_\theta f_\theta(\mathbf{x}')\|} \tag{2}$$

which has the advantage of being bounded (in $[-1, 1]$), thus expressing similarity in a usual meaning.

## 2.2 Properties for vanilla neural networks

Intuitively, inputs that are similar from the network perspective should produce similar outputs; we can check that $k_\theta^C$ is a good similarity measure in this respect (all proofs are deferred to the supplementary materials):

**Theorem 1.** *For any real-valued neural network $f_\theta$ whose last layer is a linear layer (without any parameter sharing) or a standard activation function thereof (sigmoid, tanh, ReLU...), and for any inputs $\mathbf{x}$ and $\mathbf{x}'$,*

$$\nabla_\theta f_\theta(\mathbf{x}) = \nabla_\theta f_\theta(\mathbf{x}') \implies f_\theta(\mathbf{x}) = f_\theta(\mathbf{x}').$$

**Corollary 1.** *Under the same assumptions, for any inputs $\mathbf{x}$ and $\mathbf{x}'$,*

$$k_\theta^C(\mathbf{x}, \mathbf{x}') = 1 \implies \nabla_\theta f_\theta(\mathbf{x}) = \nabla_\theta f_\theta(\mathbf{x}'),$$
$$\text{hence} \quad k_\theta^C(\mathbf{x}, \mathbf{x}') = 1 \implies f_\theta(\mathbf{x}) = f_\theta(\mathbf{x}').$$

Furthermore,

**Theorem 2.** *For any real-valued neural network $f_\theta$ without parameter sharing, if $\nabla_\theta f_\theta(\mathbf{x}) = \nabla_\theta f_\theta(\mathbf{x}')$ for two inputs $\mathbf{x}, \mathbf{x}'$, then all useful activities computed when processing $\mathbf{x}$ are equal to the ones obtained when processing $\mathbf{x}'$.*

We name *useful* activities all activities $a_i(\mathbf{x})$ whose variation would have an impact on the output, *i.e.* all the ones satisfying $\frac{df_\theta(\mathbf{x})}{da_i} \neq 0$. This condition is typically not satisfied when the activity is negative and followed by a ReLU, or when it is multiplied by a 0 weight, or when all its contributions to the output cancel one another (*e.g.*, a sum of two neurons with opposite weights: $f_\theta(\mathbf{x}) = \sigma(a_i(\mathbf{x})) - \sigma(a_i(\mathbf{x}))$).

**Link with the *perceptual loss*** For a vanilla network without parameter sharing, the gradient $\nabla_\theta f_\theta(\mathbf{x})$ is a list of coefficients $\nabla_{w_i^j} f_\theta(\mathbf{x}) = \frac{df_\theta(\mathbf{x})}{db_j} a_i(\mathbf{x})$, where $w_i^j$ is the parameter-factor that multiplies the input activation $a_i(\mathbf{x})$ in neuron $j$, and of coefficients $\nabla_{b_j} f_\theta(\mathbf{x}) = \frac{df_\theta(\mathbf{x})}{db_j}$ for neuron biases, which we will consider as standard parameters $b_j = w_0^j$ that act on a constant activation $a_0(\mathbf{x}) = 1$, yielding $\nabla_{w_0^j} f_\theta(\mathbf{x}) = \frac{df_\theta(\mathbf{x})}{db_j} a_0(\mathbf{x})$. Thus the gradient $\nabla_\theta f_\theta(\mathbf{x})$ can be seen as a list of all activation values $a_i(\mathbf{x})$ multiplied by the potential impact on the output $f_\theta(\mathbf{x})$ of the neurons $j$ using them, *i.e.* $\frac{df_\theta(\mathbf{x})}{db_j}$. Each activation appears in this list as many times as it is fed to different neurons. The similarity between two inputs then rewrites:

$$k_\theta^I(\mathbf{x}, \mathbf{x}') = \sum_{\text{activities } i} \lambda_i(\mathbf{x}, \mathbf{x}') \, a_i(\mathbf{x}) \, a_i(\mathbf{x}') \quad \text{where} \quad \lambda_i(\mathbf{x}, \mathbf{x}') = \sum_{\text{neuron } j \text{ using } a_i} \frac{df_\theta(\mathbf{x})}{db_j} \frac{df_\theta(\mathbf{x}')}{db_j}$$

are data-dependent importance weights. Such weighting schemes on activation units naturally arise when expressing intrinsic quantities; the use of natural gradients would bring invariance to re-parameterization [16, 17]. On the other hand, the inner product related to the perceptual loss would be

$$\sum_{\text{activities } i \neq 0} \lambda_{\text{layer}(i)} \, a_i(\mathbf{x}) \, a_i(\mathbf{x}')$$

for some arbitrary fixed layer-dependent weights $\lambda_{\text{layer}(i)}$.

## 2.3 Properties for parameter-sharing networks

When sharing weights, as in convolutional networks, the gradient $\nabla_\theta f_\theta(\mathbf{x})$ is made of the same coefficients (impact-weighted activations) but summed over shared parameters. Denoting by $\mathcal{S}(i)$ the set of (neuron, input activity) pairs where the parameter $w_i$ is involved,

$$k_\theta^I(\mathbf{x}, \mathbf{x}') = \sum_{\text{params } i} \left( \sum_{(j,k) \in \mathcal{S}_i} a_k(\mathbf{x}) \frac{df_\theta(\mathbf{x})}{db_j} \right) \left( \sum_{(j,k) \in \mathcal{S}_i} a_k(\mathbf{x}') \frac{df_\theta(\mathbf{x}')}{db_j} \right)$$

Thus, in convolutional networks, $k_\theta^I$ similarity does not imply similarity of first layer activations anymore, but only of their (impact-weighted) spatial average. More generally, any invariance

introduced by a weight sharing scheme in an architecture will be reflected in the similarity measure $k_\theta^I$, which is expected as $k_\theta^I$ was defined as the input similarity *from the neural network perspective*.

Note that this type of objects was recently studied from an optimization viewpoint under the name of Neural Tangent Kernel [9, 1] in the infinite layer width limit.

## 3 Higher output dimension

Let us now study the more complex case where $f_\theta(\mathbf{x})$ is a vector $\left(f_\theta^i(\mathbf{x})\right)_{i \in [1,d]}$ in $\mathbb{R}^d$ with $d > 1$. Under a mild hypothesis on the network (output expressivity), always satisfied unless specially designed not to:

**Theorem 3.** *The optimal parameter change $\delta\theta$ to push $f_\theta(\mathbf{x})$ in a direction $\mathbf{v} \in \mathbb{R}^d$ (with a force $\varepsilon \in \mathbb{R}$), i.e. such that $f_{\theta+\delta\theta}(\mathbf{x}) - f_\theta(\mathbf{x}) = \varepsilon\mathbf{v}$, induces at any other point $\mathbf{x}'$ the following output variation:*

$$f_{\theta+\delta\theta}(\mathbf{x}') - f_\theta(\mathbf{x}') = \varepsilon\, K_\theta(\mathbf{x}',\mathbf{x})\, K_\theta(\mathbf{x},\mathbf{x})^{-1}\, \mathbf{v}\, +\, O(\varepsilon^2) \tag{3}$$

*where the $d \times d$ kernel matrix $K_\theta(\mathbf{x}',\mathbf{x})$ is defined by $K_\theta^{ij}(\mathbf{x}',\mathbf{x}) = \nabla_\theta f_\theta^i(\mathbf{x}') \cdot \nabla_\theta f_\theta^j(\mathbf{x})$.*

The similarity kernel is now a matrix and not just a single value, as it describes the relation between moves $\mathbf{v} \in \mathbb{R}^d$. Note that these matrices $K_\theta$ are only $d \times d$ where $d$ is the output dimension. They are thus generally small and easy to manipulate or inverse.

**Normalized similarity matrix** The unitless symmetrized, normalized version of the kernel (3) is:

$$K_\theta^C(\mathbf{x},\mathbf{x}') \;=\; K_\theta(\mathbf{x},\mathbf{x})^{-1/2}\, K_\theta(\mathbf{x},\mathbf{x}')\, K_\theta(\mathbf{x}',\mathbf{x}')^{-1/2}\;. \tag{4}$$

It has the following properties: its coefficients are bounded, in $[-1, 1]$; its trace is at most $d$; its (Frobenius) norm is at most $\sqrt{d}$; self-similarity is identity: $\forall \mathbf{x},\; K_\theta^C(\mathbf{x},\mathbf{x}) = \mathrm{Id}$; the kernel is symmetric, in the sense that $K_\theta^C(\mathbf{x}',\mathbf{x}) = K_\theta^C(\mathbf{x},\mathbf{x}')^T$.

**Similarity in a single value** To summarize the similarity matrix $K_\theta^C(\mathbf{x},\mathbf{x}')$ into a single real value in $[-1, 1]$, we consider:

$$k_\theta^C(\mathbf{x},\mathbf{x}') \;=\; \frac{1}{d}\,\mathrm{Tr}\, K_\theta^C(\mathbf{x},\mathbf{x}')\;. \tag{5}$$

It can be shown indeed that if $k_\theta^C(\mathbf{x},\mathbf{x}')$ is close to 1, then $K_\theta^C(\mathbf{x},\mathbf{x}')$ is close to $\mathrm{Id}$, and reciprocally. See the supplementary materials for more details and a discussion about the links between $\frac{1}{d}\,\mathrm{Tr}\, K_\theta^C(\mathbf{x},\mathbf{x}')$ and $\left\|K_\theta^C(\mathbf{x},\mathbf{x}') - \mathrm{Id}\right\|_F$.

**Metrics on output: rotation invariance** Similarity in $\mathbb{R}^d$ might be richer than just estimating distances in $L^2$ norm. For instance, for our 2D image registration task, the network could be known (or desired) to be equivariant to rotations. The similarity between two output variations $\mathbf{v}$ and $\mathbf{v}'$ can be made rotation-invariant by applying the rotation that best aligns $\mathbf{v}$ and $\mathbf{v}'$ beforehand. This can actually be easily computed in closed form and yields:

$$k_\theta^{C,\mathrm{rot}}(\mathbf{x},\mathbf{x}') \;=\; \frac{1}{2}\sqrt{\left\|K_\theta^C(\mathbf{x},\mathbf{x}')\right\|_F^2 + 2\det K_\theta^C(\mathbf{x},\mathbf{x}')}\;.$$

Note that other metrics are possible in the output space. For instance, the loss metric quantifies the norm of a move $\mathbf{v}$ by its impact on the loss $\frac{dL(y)}{dy}\big|_{f_\theta(\mathbf{x})}(\mathbf{v})$. It has a particular meaning though, and is not always relevant, *e.g.* in the noisy label case seen in Section 5.

**The case of classification tasks** When the output of the network is a probability distribution $p_{\theta,\mathbf{x}}(c)$, over a finite number of given classes $c$ for example, it is natural from an information theoretic point of view to rather consider $f_\theta^c(\mathbf{x}) = -\log p_{\theta,\mathbf{x}}(c)$. This is actually the quantities computed in the pre-softmax layer from which common practice directly computes the cross-entropy loss.

It turns out that the $L^2$ norm of variations $\delta f$ in this space naturally corresponds to the Fisher information metric, which quantifies the impact of parameter variations $\delta\theta$ on the output probability $p_{\theta,\mathbf{x}}$, as $\mathrm{KL}(p_{\theta,\mathbf{x}} || p_{\theta+\delta\theta,\mathbf{x}})$. The matrices $K_\theta(\mathbf{x},\mathbf{x}) = \left(\nabla_\theta f_\theta^c(\mathbf{x}) \cdot \nabla_\theta f_\theta^{c'}(\mathbf{x})\right)_{c,c'}$ and $F_{\theta,\mathbf{x}} = \mathbb{E}_c\left[\nabla_\theta f_\theta^c(\mathbf{x})\, \nabla_\theta f_\theta^c(\mathbf{x})^T\right]$ are indeed to each other what correlation is to covariance. Thus the quantities defined in Equation (5) already take into account information geometry when applied to the pre-softmax layer, and do not need supplementary metric adjustment.

**Faster setup for classification tasks with many classes** In a classification task in $d$ classes with large $d$, the computation of $d \times d$ matrices may be prohibitive. As a workaround, for a given input training sample $\mathbf{x}$, the classification task can be seen as a binary one (the right label $c_R$ *vs.* the other ones), in which case the $d$ outputs of the neural network can be accordingly combined in a single real value. The 1D similarity measure can then be used to compare any training samples of the same class.

When making statistics on similarity values $\mathbb{E}_{\mathbf{x}'}\left[k_\theta^C(\mathbf{x}, \mathbf{x}')\right]$, another possible task binarization approach is to sample an adversary class $c_A$ along with $\mathbf{x}'$, and hence consider $\nabla_\theta f_\theta^{c_R}(\mathbf{x}) - \nabla_\theta f_\theta^{c_A}(\mathbf{x})$. Both approaches will lead to similar results in the Enforcing Similarity section in the supplementary materials.

# 4 Estimating density

In this section, we use similarity to estimate input neighborhoods and perform statistics on them.

## 4.1 Estimating the number of neighbors

Given a point $\mathbf{x}$, how many samples $\mathbf{x}'$ are similar to $\mathbf{x}$ according to the network? This can be measured by computing $k_\theta^C(\mathbf{x}, \mathbf{x}')$ for all $\mathbf{x}'$ and picking the closest ones, *i.e. e.g.* the $\mathbf{x}'$ such that $k_\theta^C(\mathbf{x}, \mathbf{x}') \geqslant 0.9$. More generally, for any data point $\mathbf{x}$, the histogram of the similarity $k_\theta^C(\mathbf{x}, \mathbf{x}')$ over all $\mathbf{x}'$ in the dataset (or a representative subset thereof) can be drawn, and turned into an estimate of the number of neighbors of $\mathbf{x}$. To do this, several types of estimates are possible:

- hard-thresholding, for a given threshold $\tau \in [0, 1]$: $\qquad N_\tau(\mathbf{x}) = \sum_{\mathbf{x}'} \mathbb{1}_{k_\theta^C(\mathbf{x}, \mathbf{x}') \geqslant \tau}$
- soft estimate: $\qquad\qquad\qquad\qquad\qquad\qquad N_S(\mathbf{x}) = \sum_{\mathbf{x}'} k_\theta^C(\mathbf{x}, \mathbf{x}')$
- less-soft positive-only estimate ($\alpha > 0$): $\qquad N_\alpha^+(\mathbf{x}) = \sum_{\mathbf{x}'} \mathbb{1}_{k_\theta^C(\mathbf{x}, \mathbf{x}') > 0} \, k_\theta^C(\mathbf{x}, \mathbf{x}')^\alpha$

In practice we observe that $k_\theta^C$ is very rarely negative, and thus the soft estimate $N_S$ can be justified as an average of the hard-thresholding estimate $N_\tau$ over all possible thresholds $\tau$:

$$\int_{\tau=0}^1 N_\tau(\mathbf{x}) d\tau = \sum_{\mathbf{x}'} \int_{\tau=0}^1 \mathbb{1}_{k_\theta^C(\mathbf{x}, \mathbf{x}') \geqslant \tau} \, d\tau = \sum_{\mathbf{x}'} k_\theta^C(\mathbf{x}, \mathbf{x}') \, \mathbb{1}_{k_\theta^C(\mathbf{x}, \mathbf{x}') \geqslant 0} = N_1^+(\mathbf{x}) \simeq N_S(\mathbf{x})$$

## 4.2 Low complexity of the soft estimate $N_S(\mathbf{x})$

The soft estimate $N_S(\mathbf{x})$ is rewritable as:

$$\sum_{\mathbf{x}'} k_\theta^C(\mathbf{x}, \mathbf{x}') = \sum_{\mathbf{x}'} \frac{\nabla_\theta f_\theta(\mathbf{x})}{\|\nabla_\theta f_\theta(\mathbf{x})\|} \cdot \frac{\nabla_\theta f_\theta(\mathbf{x}')}{\|\nabla_\theta f_\theta(\mathbf{x}')\|} = \frac{\nabla_\theta f_\theta(\mathbf{x})}{\|\nabla_\theta f_\theta(\mathbf{x})\|} \cdot \mathbf{g} \quad \text{with} \quad \mathbf{g} = \sum_{\mathbf{x}'} \frac{\nabla_\theta f_\theta(\mathbf{x}')}{\|\nabla_\theta f_\theta(\mathbf{x}')\|}$$

and consequently $N_S(\mathbf{x})$ can be computed jointly for all $\mathbf{x}$ in linear time $O(|\mathcal{D}|p)$ in the dataset size $|\mathcal{D}|$ and in the number of parameters $p$, in just two passes over the dataset, when the output dimension is 1. For higher output dimensions $d$, a similar trick can be used and the complexity becomes $O(|\mathcal{D}|d^2 p)$. For classification tasks with a large number $d$ of classes, the complexity can be reduced to $O(|\mathcal{D}|p)$ through an approximation consisting in binarizing the task (*c.f.* end of Section 3).

## 4.3 Test of the various estimators

In order to rapidly test the behavior of all possible estimators, we applied them to a toy problem where the network's goal is to predict a sinusoid. To change the difficulty of the problem, we vary its frequency, while keeping the number of samples constant. More details and results of the toy problem are in the supplementary materials. Fig. 2 shows for each estimator (with different parameters when relevant), the result of their neighbor count estimation. When the frequency $f$ of the sinusoid to predict increases, the number of neighbors decreases in $\frac{1}{f}$ for every estimator. This aligns with our intuition that as the problem gets harder, the network needs to distinguish input samples more to achieve a good performance, thus the amount of neighbors is lower. In particular we observe that the proposed $N_S(\mathbf{x})$ estimator behaves well, thus we will use that one in bigger studies requiring an efficient estimator.

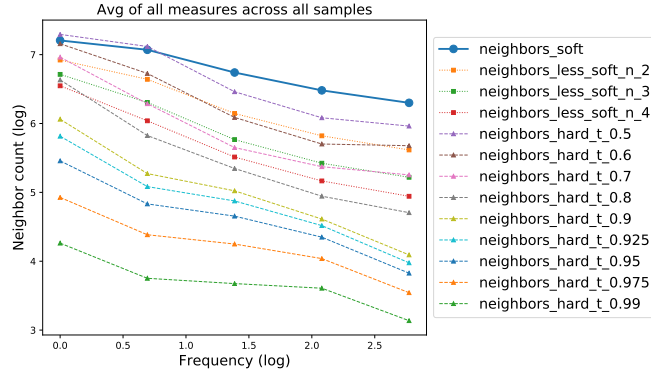

Figure 2: Density estimation using the various approaches (log scale). All approaches behave similarly and show good results, except the ones with extreme thresholds.

## 4.4 Further potential uses for fitness estimation

When the number of neighbors of a training point $\mathbf{x}$ is very low, the network is able to set any label to $\mathbf{x}$, as this won't interfere with other points, by definition of our similarity criterion $k_\theta(\mathbf{x}, \mathbf{x}')$. This is thus a typical overfit case, where the network can learn by heart a label associated to a particular, isolated point.

On the opposite, when the set of neighbors of $\mathbf{x}$ is a large fraction of the dataset, comprising varied elements, by definition of $k_\theta(\mathbf{x}, \mathbf{x}')$ the network is not able to distinguish them, and consequently it can only provide a common output for all of them. Therefore it might not be able to express variety enough, which would be a typical underfit case.

The quality of fit can thus be observed by monitoring the number of neighbors together with the variance of the desired labels in the neighborhoods (to distinguish underfit from just high density).

**Prediction uncertainty** A measure of the uncertainty of a prediction $f_\theta(\mathbf{x})$ could be to check how easy it would have been to obtain another value during training, without disturbing the training of other points. A given change $\mathbf{v}$ of $f_\theta(\mathbf{x})$ induces changes $\frac{k_\theta^I(\mathbf{x}, \mathbf{x}')}{\|\nabla_\theta f_\theta(\mathbf{x})\|^2}\mathbf{v}$ over other points $\mathbf{x}'$ of the dataset, creating a total $L^1$ disturbance $\sum_{\mathbf{x}'} \|\frac{k_\theta^I(\mathbf{x}, \mathbf{x}')}{\|\nabla_\theta f_\theta(\mathbf{x})\|^2}\mathbf{v}\|$. The uncertainty factor would then be the norm of $\mathbf{v}$ affordable within a disturbance level, and quickly approximable as $\frac{\|\nabla_\theta f_\theta(\mathbf{x})\|^2}{\sum_{\mathbf{x}'} k_\theta^I(\mathbf{x}, \mathbf{x}')}$.

## 5 Dataset self-denoising

### 5.1 Motivation: example of remote sensing image registration with noisy labels

In remote sensing imagery, data is abundant but noisy [14]. For instance RGB satellite images and binary cadaster maps (delineating buildings) are numerous but badly aligned for various reasons (annotation mistakes, atmosphere disturbance, elevation variations...). In a recent preliminary work [6], we tackled the task of automatically registering these two types of images together with neural networks, training on a dataset [13] with noisy annotations from OSM[18], and hoping the network would be able to learn from such a dataset of imperfect alignments. Learning with noisy labels is indeed an active topic of research [21, 15, 12].

For this, we designed an iterative approach: train, then use the outputs of the network on the training set to re-align it; repeat (for 3 iterations). The results were surprisingly good, yielding far better alignments than the ground truth it learned from, both qualitatively (Figure 3) and quantitatively (Figure 4, obtained on manually-aligned data): the median registration error dropped from 18 pixels to 3.5 pixels, which is the best score one could hope for, given intrinsic ambiguities in such registration task. To check that this performance was not due to a subset of the training data that would be perfectly aligned, we added noise to the ground truth and re-trained from it: the new results were about as good again (dashed lines). Thus the network did learn almost perfectly just from noisy labels.

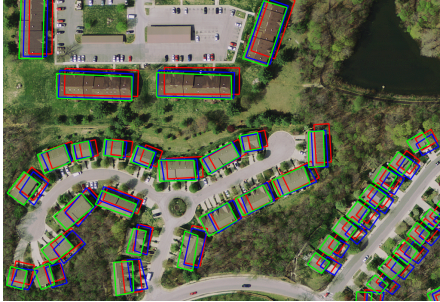

Figure 3: Qualitative alignment results [6] on a crop of bloomington22 from the Inria dataset [13]. Red: initial dataset annotations; blue: aligned annotations round 1; green: aligned annotations round 2.

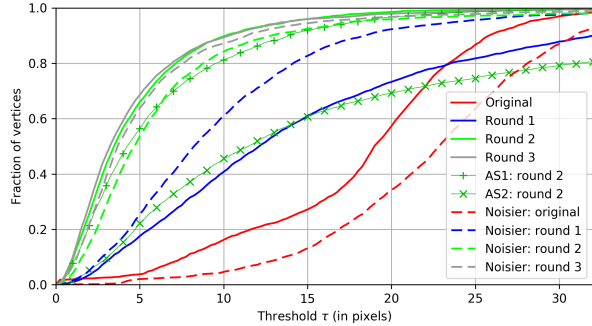

Figure 4: Accuracy cumulative distributions [6] measured with the manually-aligned annotations of bloomington22 [13]. Read as: fraction of image pixels whose registration error is less than threshold $\tau$.

An explanation for this self-denoising phenomenon is proposed in [11] as follows. Let us consider a regression task, with a $L^2$ loss, and where true labels $y$ were altered with i.i.d. noise $\varepsilon$ of variance $v$. Suppose a same input $\mathbf{x}$ appears $n$ times in the training set, thus with $n$ different labels $\widetilde{y}_i = y + \varepsilon_i$. The network can only output the same prediction for all these $n$ cases (since the input is the same), and the best option, considering the $L^2$ loss, is to predict the average $\frac{1}{n}\sum_i \widetilde{y}_i$, whose distance to the true label $y$ is $O(\frac{v}{\sqrt{n}})$. Thus a denoising effect by a factor $\sqrt{n}$ can be observed. However, the exact same point $\mathbf{x}$ is not likely to appear several times in a dataset (with different labels). Rather, relatively *similar* points may appear, and the amplitude of the self-denoising effect will be a function of their number. Here, the similarity should reflect the neural network perception (similar inputs yield the same output) and not an *a priori* norm chosen on the input space.

## 5.2 Similarity experimentally observed between patches

We studied the multi-round training scheme of [6] by applying our similarity measure to a sampling of input patches of the training dataset for one network per round. The principle of the multiple round training scheme is to reduce the noise of the annotations, obtaining aligned annotations in the end (more details in the supplementary materials). For a certain input patch, we computed its similarity with all the other patches for the 3 networks. With those similarities we can compute the nearest neighbors of that patch, see Fig. 5. The input patch is of a suburb area with sparse houses and individual trees. The closest neighbors look similar as they usually feature the same types of buildings, building arrangement and vegetation. However sometimes the network sees a patch as similar when it is not clear from our point of view (for example patches with large buildings).

For more in-depth results, we computed the histogram of similarities for the same patch, see Fig. 6. We observe that round 2 shows different neighborhood statistics, in that the patch is closer to all other patches than in other rounds. We observe the same behavior in 19 other input patches (see suppl. materials). An hypothesis for this phenomenon is that the average gradient was not 0 at the end of that training round (due to optimization convergence issues, e.g.), which would shift all similarity histograms by a same value.

Qualitatively, for patches randomly sampled, their similarity histograms tend to be approximately symmetric in round 2, but with a longer left tail in round 1 and a longer right tail in round 3. Neighborhoods thus seem to change across the rounds, with fewer and fewer close points (if removing the global histogram shift in round 2). A possible interpretation is that this would reflect an increasing ability of the network to distinguish between different patches, with finer features in later training rounds.

## 5.3 Comparison to the *perceptual loss*

We compare our approach to the *perceptual loss* on a nearest neighbor retrieval task. We notice that the *perceptual loss* sometimes performs reasonably well, but often not. For instance, we show in Fig. 7 the closest neighbors to a structured residential area image, for the *perceptual loss* (first row: not making sense) and for our similarity measure (second row: similar areas).

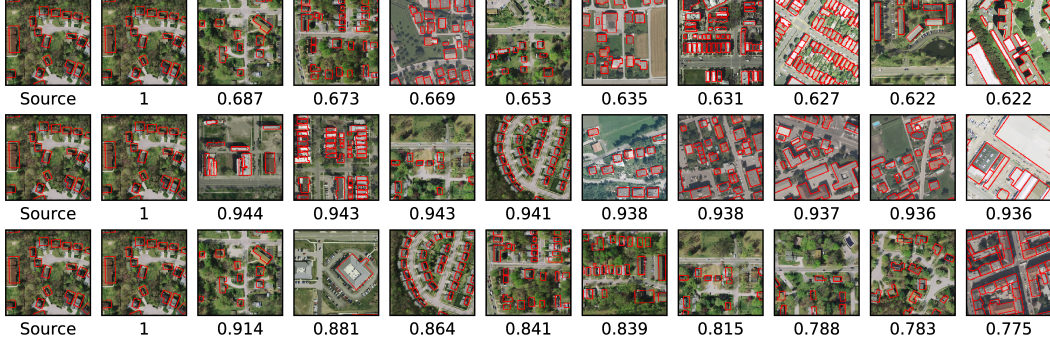

Figure 5: Example of nearest neighbors for a patch. Each line corresponds to a round. Each patch has its similarity written under it.

(a) Round 1      (b) Round 2      (c) Round 3

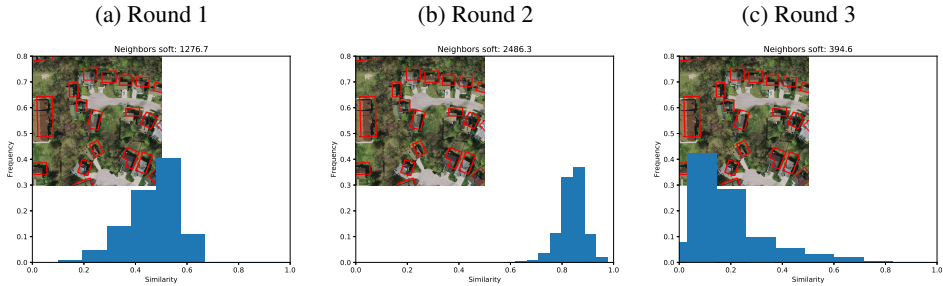

Figure 6: Histograms of similarities for one patch across rounds.

## 5.4 From similarity statistics to self-denoising effect estimation

We now show how such similarity experimental computations can be used to solve the initial problem of Section 5, by explicitly turning similarity statistics into a quantification of the self-denoising effect.

Let us denote by $y_i$ the true (unknown) label for input $\mathbf{x}_i$, by $\widetilde{y}_i$ the noisy label given in the dataset, and by $\widehat{y}_i = f_\theta(\mathbf{x}_i)$ the label predicted by the network. We will denote the (unknown) noise by $\varepsilon_i = \widetilde{y}_i - y_i$ and assume it is centered and i.i.d., with finite variance $\sigma_\varepsilon$. The training criterion is $E(\theta) = \sum_j \|\widehat{y}_j - \widetilde{y}_j\|^2$. At convergence, the training leads to a local optimum of the energy landscape: $\nabla_\theta E = 0$, that is, $\sum_j (\widehat{y}_j - \widetilde{y}_j) \nabla_\theta \widehat{y}_j = 0$. Let's choose any sample $i$ and multiply by $\nabla_\theta \widehat{y}_i$: using $k_\theta^I(\mathbf{x}_i, \mathbf{x}_j) = \nabla_\theta \widehat{y}_i . \nabla_\theta \widehat{y}_j$, we get:

$$\sum_j (\widehat{y}_j - \widetilde{y}_j)\, k_\theta^I(\mathbf{x}_j, \mathbf{x}_i) = 0.$$

Let us denote by $k_\theta^{IN}(\mathbf{x}_j, \mathbf{x}_i) = k_\theta^I(\mathbf{x}_j, \mathbf{x}_i)\big(\sum_j k_\theta^I(\mathbf{x}_j, \mathbf{x}_i)\big)^{-1}$ the column-normalized kernel, and by $\mathbb{E}_k[a] = \sum_j a_j\, k_\theta^{IN}(\mathbf{x}_j, \mathbf{x}_i)$ the mean value of $a$ in the neighborhood of $i$, that is, the weighted

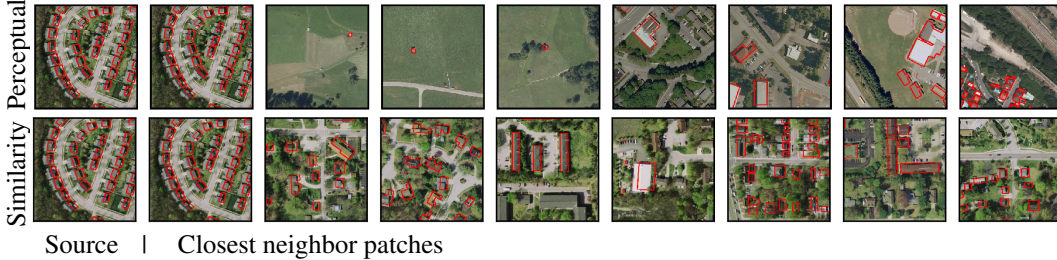

Source  |  Closest neighbor patches

Figure 7: Closest neighbors to the leftmost patch, using the *perceptual loss* (first row) and our similarity definition (second row).

average of the $a_j$ with weights $k_\theta^I(\mathbf{x}_j, \mathbf{x}_i)$ normalized to sum up to 1. This is actually a kernel regression, in the spirit of Parzen-Rosenblatt window estimators. Then the previous property can be rewritten as $\mathbb{E}_k[\widehat{y}] = \mathbb{E}_k[\widetilde{y}]$. As $\mathbb{E}_k[\widetilde{y}] = \mathbb{E}_k[y] + \mathbb{E}_k[\varepsilon]$, this yields:

$$\widehat{y}_i - \mathbb{E}_k[y] = \mathbb{E}_k[\varepsilon] + (\widehat{y}_i - \mathbb{E}_k[\widehat{y}])$$

*i.e.* the difference between the predicted $\widehat{y}_i$ and the average of the true labels in the neighborhood of $i$ is equal to the average of the noise in the neighborhood of $i$, up to the deviation of the prediction $\widehat{y}_i$ from the average prediction in its neighborhood.

We want to bound the error $\|\widehat{y}_i - \mathbb{E}_k[y]\|$ without knowing neither the true labels $y$ nor the noise $\varepsilon$. One can show that $\mathbb{E}_k[\varepsilon] \propto \mathrm{var}_\varepsilon(\mathbb{E}_k[\varepsilon])^{1/2} = \sigma_\varepsilon \|k_\theta^{IN}(\cdot, \mathbf{x}_i)\|_{L2}$. The denoising factor is thus the similarity kernel norm $\|k_\theta^{IN}(\cdot, \mathbf{x}_i)\|_{L2}$, which is between $1/\sqrt{N}$ and 1, depending on the neighborhood quality. It is $1/\sqrt{N}$ when all $N$ data points are identical, i.e. all satisfying $k_\theta^C(\mathbf{x}_i, \mathbf{x}_j) = 1$. On the other extreme, this factor is 1 when all points are independent: $k_\theta^I(\mathbf{x}_i, \mathbf{x}_j) = 0 \quad \forall i \neq j$. This way we extend *noise2noise* [11] to real datasets with non-identical inputs.

In our remote sensing experiment, we estimate this way a denoising factor of 0.02, consistent across all training rounds and inputs ($\pm 10\%$), implying that each training round contributed equally to denoising the labels. This is confirmed by Fig. 4, which shows the error steadily decreasing, on a control test where true labels are known. The shift $(\widehat{y}_i - \mathbb{E}_k[\widehat{y}])$ on the other hand can be directly estimated given the network prediction. In our case, it is 4.4px on average, which is close to the observed median error for the last round in Fig. 4. It is largely input-dependent, with variance 3.2px, which is reflected by the spread distribution of errors in Fig. 4. This input-dependent shift thus provides a hint about prediction reliability.

It is also possible to bound $(\widehat{y}_i - \mathbb{E}_k[\widehat{y}]) = \mathbb{E}_k[\widehat{y}_i - \widehat{y}]$ using only similarity information (without predictions $\widehat{y}$). Theorem 1 implies that the application: $\frac{\nabla_\theta f_\theta(\mathbf{x})}{\|\nabla_\theta f_\theta(\mathbf{x})\|} \mapsto f_\theta(\mathbf{x})$ is well-defined, and it can actually be shown to be Lipschitz with a network-dependent constant (under mild hypotheses). Thus

$$\|f_\theta(\mathbf{x}) - f_\theta(\mathbf{x}')\| \leqslant C \left\| \frac{\nabla_\theta f_\theta(\mathbf{x})}{\|\nabla_\theta f_\theta(\mathbf{x})\|} - \frac{\nabla_\theta f_\theta(\mathbf{x}')}{\|\nabla_\theta f_\theta(\mathbf{x}')\|} \right\| = \sqrt{2}C\sqrt{1 - k_\theta^C(\mathbf{x}, \mathbf{x}')} \,,$$

yielding $\|\widehat{y}_i - \widehat{y}_j\| \leqslant \sqrt{2}C\sqrt{1 - k_\theta^C(\mathbf{x}_i, \mathbf{x}_j)}$ and thus $\left| \mathbb{E}_k[\widehat{y}_i - \widehat{y}] \right| \leqslant \sqrt{2}C \, \mathbb{E}_k\left[ \sqrt{1 - k_\theta^C(\mathbf{x}_i, \cdot)} \right]$.

# 6   Conclusion

We defined a proper notion of input similarity as perceived by the neural network, based on the ability of the network to distinguish the inputs. This brings a new tool to analyze trained networks, in plus of visualization tools such as grad-CAM [20]. We showed how to turn it into a density estimator, which was validated on a controlled experiment, and usable to perform fast statistics on large datasets. It opens the door to underfit/overfit/uncertainty analyses or even control during training, as it is differentiable and computable at low cost.

In the supplementary materials, we go further in that direction and show that, if two or more samples are known to be similar (from a human point of view), it is possible to incite the network, during training, to evolve in order to consider these samples as similar. We notice an associated dataset-dependent boosting effect that should be further studied along with robustness to adversarial attacks, as such training differs significantly from usual methods.

Finally, we extended *noise2noise* [11] to the case of non-identical inputs, thus expressing self-denoising effects as a function of inputs' similarities.

The code is available at `https://github.com/Lydorn/netsimilarity` .

## Acknowledgments

We thank Victor Berger and Adrien Bousseau for useful discussions. This work benefited from the support of the project EPITOME ANR-17-CE23-0009 of the French National Research Agency (ANR).

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
