[Supplementary Material · suppl_mat.pdf]

# Supplementary materials for
# Input Similarity from the Neural Network Perspective

**Guillaume Charpiat**[1]    **Nicolas Girard**[2]    **Loris Felardos**[1]    **Yuliya Tarabalka**[2,3]

[1] TAU team, INRIA Saclay, LRI, Univ. Paris-Sud
[2] TITANE team, INRIA Sophia-Antipolis, Univ. Côte d'Azur
[3] LuxCarta Technology
`firstname.lastname@inria.fr`

## 1   Foreword

**Code**   The whole code (image registration, experiments to test density estimators, enforcing similarity...) is available on the following github repository: `https://github.com/Lydorn/netsimilarity`.

**Content of this document**   This document contains further details on the article *Input Similarity from the Neural Network Perspective*, such as mathematical proofs, experimental details and further discussions.

## 2   Proofs of the properties of the 1D similarity kernel

We give here the proofs at the properties of the 1-dimensional-output similarity kernel.

### 2.1   Proof of Theorem 1

**Theorem 1.** *For any real-valued neural network $f_\theta$ whose last layer is a linear layer (without any parameter sharing) or a standard activation function thereof (sigmoid, tanh, ReLU...), and for any inputs $\mathbf{x}$ and $\mathbf{x}'$,*

$$\nabla_\theta f_\theta(\mathbf{x}) = \nabla_\theta f_\theta(\mathbf{x}') \implies f_\theta(\mathbf{x}) = f_\theta(\mathbf{x}')\,.$$

*Proof.* If the last layer is linear, the output is of the form $f_\theta(\mathbf{x}) = \sum_i w_i a_i(\mathbf{x}) + b$, where $w_i$ and $b$ are parameters in $\mathbb{R}$ and $a_i(\mathbf{x})$ activities from previous layers. The gradient $\nabla_\theta f_\theta(\mathbf{x})$ contains in particular as coefficients the derivatives $\frac{df_\theta(\mathbf{x})}{dw_i} = a_i(\mathbf{x})$. Thus $\nabla_\theta f_\theta(\mathbf{x}) = \nabla_\theta f_\theta(\mathbf{x}') \implies a_i(\mathbf{x}) = a_i(\mathbf{x}')\ \forall i$ in the last layer. The outputs can be then rebuilt: $f_\theta(\mathbf{x}) = \sum_i w_i a_i(\mathbf{x}) + b = \sum_i w_i a_i(\mathbf{x}') + b = f_\theta(\mathbf{x}')$.

If the output is of the form $f_\theta(\mathbf{x}) = \sigma(c(\mathbf{x}))$ with $c(\mathbf{x}) = \sum_i w_i a_i(\mathbf{x}) + b$, then the gradient equality implies $\frac{df_\theta(\mathbf{x})}{db} = \frac{df_\theta(\mathbf{x}')}{db}$, whose value is $\sigma'(c(\mathbf{x})) = \sigma'(c(\mathbf{x}'))$. Then, as $\sigma'(c(\mathbf{x}))\, a_i(\mathbf{x}) = \frac{df_\theta(\mathbf{x})}{dw_i} = \frac{df_\theta(\mathbf{x}')}{dw_i} = \sigma'(c(\mathbf{x}'))\, a_i(\mathbf{x}')$, we can deduce $a_i(\mathbf{x}) = a_i(\mathbf{x}')$ for all $i$ provided $\sigma'(c(\mathbf{x})) \neq 0$. In that case, from these identical activities one can rebuild identical outputs. Otherwise, $\sigma'(c(\mathbf{x})) = \sigma'(c(\mathbf{x}')) = 0$, which is not possible with strictly monotonous activation functions, such as tanh or sigmoid. For ReLU, $\sigma'(c(\mathbf{x})) = 0 \implies \sigma(c(\mathbf{x})) = 0$ and thus $f_\theta(\mathbf{x}) = f_\theta(\mathbf{x}') = 0$. The same reasoning holds for other activation functions with only one flat piece (such as the ReLU negative part), *i.e.* for which the set $\sigma(\sigma'^{-1}(\{0\}))$ is a singleton.  □

## 2.2 Proof of Corollary 1

**Corollary 1.** *Under the same assumptions, for any inputs* $\mathbf{x}$ *and* $\mathbf{x}'$,

$$k_\theta^C(\mathbf{x}.\mathbf{x}') = 1 \quad \Longrightarrow \quad \nabla_\theta f_\theta(\mathbf{x}) = \nabla_\theta f_\theta(\mathbf{x}'),$$
$$\text{hence} \quad k_\theta^C(\mathbf{x}.\mathbf{x}') = 1 \quad \Longrightarrow \quad f_\theta(\mathbf{x}) = f_\theta(\mathbf{x}').$$

*Proof.* $k_\theta^C(\mathbf{x}.\mathbf{x}') = 1$ means $\frac{\nabla_\theta f_\theta(\mathbf{x})}{\|\nabla_\theta f_\theta(\mathbf{x})\|} \cdot \frac{\nabla_\theta f_\theta(\mathbf{x}')}{\|\nabla_\theta f_\theta(\mathbf{x}')\|} = 1$, which implies $\exists\, \alpha \in \mathbb{R}^*,\ \nabla_\theta f_\theta(\mathbf{x}) = \alpha\, \nabla_\theta f_\theta(\mathbf{x}')$. We need to show that $\alpha = 1$. Under the assumptions of Theorem 1, following its proof:

- either the last layer is linear, the output is of the form $f_\theta(\mathbf{x}) = \sum_i w_i a_i(\mathbf{x}) + b$, and then $\nabla_b f_\theta(\mathbf{x}) = \alpha \nabla_b f_\theta(\mathbf{x}')$ while $\frac{df_\theta(\mathbf{x})}{db} = 1$ and $\frac{df_\theta(\mathbf{x}')}{db} = 1$, hence $\alpha = 1$;

- either the output is of the form $f_\theta(\mathbf{x}) = \sigma(c(\mathbf{x}))$ with $c(\mathbf{x}) = \sum_i w_i a_i(\mathbf{x}) + b$, and then $\sigma'(c(\mathbf{x})) = \nabla_b f_\theta(\mathbf{x}) = \alpha \nabla_b f_\theta(\mathbf{x}') = \alpha\, \sigma'(c(\mathbf{x}'))$, while, for any $i$, $\sigma'(c(\mathbf{x}))\, a_i(\mathbf{x}) = \frac{df_\theta(\mathbf{x})}{dw_i} = \alpha \frac{df_\theta(\mathbf{x}')}{dw_i} = \alpha\, \sigma'(c(\mathbf{x}'))\, a_i(\mathbf{x}')$. Thus, supposing $\sigma'(c(\mathbf{x})) \neq 0$, we obtain $a_i(\mathbf{x}) = a_i(\mathbf{x}')\ \forall i$, and thus we can rebuild from the activities $c(\mathbf{x}) = c(\mathbf{x}')$, from which $\sigma'(c(\mathbf{x})) = \sigma'(c(\mathbf{x}'))$ and thus $\alpha = 1$. Otherwise, $\sigma'(c(\mathbf{x})) = \sigma'(c(\mathbf{x}')) = 0$ and the two full gradients $\nabla_\theta f_\theta(\mathbf{x})$ and $\nabla_\theta f_\theta(\mathbf{x}')$ are 0 and thus equal.

The conditions for $k_\theta^C(\mathbf{x}.\mathbf{x}') = 1 \implies \nabla_\theta f_\theta(\mathbf{x}) = \nabla_\theta f_\theta(\mathbf{x}')$ to hold are actually much weaker: it is sufficient that in the whole network architecture there exists *one* useful neuron (in the sense of the next paragraph) of that type (so called *linear* but actually affine). $\qquad\square$

## 2.3 Proof of Theorem 2

**Theorem 2.** *For any real-valued neural network* $f_\theta$ *without parameter sharing, if* $\nabla_\theta f_\theta(\mathbf{x}) = \nabla_\theta f_\theta(\mathbf{x}')$ *for two inputs* $\mathbf{x}, \mathbf{x}'$, *then all useful activities computed when processing* $\mathbf{x}$ *are equal to the ones obtained when processing* $\mathbf{x}'$.

We name *useful* activities all activities whose variation would have an impact on the output, *i.e.* all the ones satisfying $\frac{df_\theta(\mathbf{x})}{da_i} \neq 0$. This condition is typically not satisfied when the activity is multiplied by 0, *i.e.* $w_i = 0$, or when it is negative and followed by a ReLU, or when all its contributions to the output annihilate together (*e.g.*, a sum of two neurons with opposite weights: $f_\theta(\mathbf{x}) = \sigma(a_i(\mathbf{x})) - \sigma(a_i(\mathbf{x}))$).

*Proof.* Let $a_i(\mathbf{x})$ be a useful activity (for $\mathbf{x}$). It is fed to at least one useful neuron, whose pre-activation output is of the form $c(\mathbf{x}) = \sum_i w_i a_i(\mathbf{x}) + b$. Then $\frac{df_\theta(\mathbf{x})}{db} = \frac{df_\theta(\mathbf{x})}{dc} \neq 0$ (the output of the neuron is useful), and $\frac{df_\theta(\mathbf{x})}{dw_i} = \frac{df_\theta(\mathbf{x})}{db} a_i(\mathbf{x})$. From the gradient equality, $a_i(\mathbf{x}) = \frac{df_\theta(\mathbf{x})}{dw_i} / \frac{df_\theta(\mathbf{x})}{db} = \frac{df_\theta(\mathbf{x}')}{dw_i} / \frac{df_\theta(\mathbf{x}')}{db} = a_i(\mathbf{x}')$. $\qquad\square$

# 3 Higher output dimension

We expand here all the mathematical aspects of the homonymous section of the article.

## 3.1 Derivation

Let us now study the case where $f_\theta(\mathbf{x})$ is a vector in $\mathbb{R}^d$ with $d > 1$.

The optimal parameter change $\delta\theta$ to push $f_\theta(\mathbf{x})$ in a direction $\mathbf{v}$ (with a force $\varepsilon$) is less straightforward to obtain. First, one can define as many gradients as output coordinates: $\nabla_\theta f_\theta^i(\mathbf{x})$, for $i \in [\![1, d]\!]$.

This family of gradients can be shown to be linearly independent, unless the architecture of the network is specifically built not to. If for instance each output coordinate has its own bias parameter, *i.e.* writes in the form $f_\theta^i(\mathbf{x}) = b_i + g_\theta(\mathbf{x})$ or $\sigma(b_i + g_\theta(\mathbf{x}))$ with a strictly monotonous activation function $\sigma$, then the derivative w.r.t. $b_i$ will be 1 (or $\sigma'$) only in the $i$-th gradient and 0 in the other ones.

Thus the $j$-th gradient contains in particular the subvector $(\frac{df^j}{db_i})_i = (\delta_{i=j})_i$, and the gradients are consequently independent. In the case where all coordinates depend on all biases, but not identically, as with a softmax, the argument stays true.

Any parameter variation $\delta\theta \in \mathbb{R}^p$ can then be uniquely decomposed as:

$$\delta\theta = \sum_{i=1}^{d} \alpha_i \nabla f_\theta^i(\mathbf{x}) \; + \; \gamma$$

where $\alpha_i \in \mathbb{R}$ and where $\gamma \in \mathbb{R}^p$ is orthogonal to all coordinate gradients. This parameter variation induces an output variation:

$$f_{\theta+\delta\theta}(\mathbf{x}) - f_\theta(\mathbf{x}) = \nabla_\theta f_\theta(\mathbf{x})\, \delta\theta + O(\|\delta\theta\|^2)$$

$$= \left( \sum_i \alpha_i \nabla_\theta f_\theta^i(\mathbf{x}) \cdot \nabla f_\theta^j(\mathbf{x}) \right)_j + 0 + O(\|\delta\theta\|^2)$$

$$= C\alpha + O(\|\alpha\|^2)$$

where $C$ is the correlation matrix of the gradients: $C_{ij} = \nabla_\theta f_\theta^i(\mathbf{x}) \cdot \nabla f_\theta^j(\mathbf{x})$. It turns out that $C$ is invertible:

$$C\alpha = 0 \implies \alpha C\alpha = 0 \implies \alpha \nabla_\theta f_\theta(\mathbf{x})\, \nabla_\theta f_\theta(\mathbf{x})\, \alpha = 0$$

$$\implies \|\nabla_\theta f_\theta(\mathbf{x})\, \alpha\|^2 = 0 \implies \sum_i \alpha_i \nabla f_\theta^i(\mathbf{x}) = 0$$

$\implies \alpha = 0$ as the $\nabla_\theta f_\theta^i(\mathbf{x})$ are linearly independent. Thus, for a desired output move in the direction $\mathbf{v}$ with amplitude $\varepsilon$, $i.e.$ $f_{\theta+\delta\theta}(\mathbf{x}) - f_\theta(\mathbf{x}) = \varepsilon\mathbf{v}$, one can compute the associated linear combination $\alpha = \varepsilon\, C^{-1}\mathbf{v}$ and thus the smallest associated parameter change $\delta\theta = \sum_i \alpha_i \nabla f_\theta^i(\mathbf{x})$.

The output variation induced at any other point $\mathbf{x}'$ by this parameter change is then:

$$f_{\theta+\delta\theta}(\mathbf{x}') - f_\theta(\mathbf{x}') = \left( \nabla_\theta f_\theta^i(\mathbf{x}') \cdot \delta\theta \right)_i + O(\|\delta\theta\|^2)$$

$$= \left( \sum_j \alpha_j \nabla_\theta f_\theta^i(\mathbf{x}') \cdot \nabla_\theta f_\theta^j(\mathbf{x}) \right)_i + O(\|\delta\theta\|^2).$$

$$= \varepsilon\, K_\theta(\mathbf{x}', \mathbf{x})\, C_\theta(\mathbf{x})^{-1}\, \mathbf{v} + O(\varepsilon^2) \tag{1}$$

where the $d \times d$ kernel matrix $K_\theta(\mathbf{x}, \mathbf{x}')$ is defined by $K_\theta^{ij}(\mathbf{x}, \mathbf{x}') = \nabla_\theta f_\theta^i(\mathbf{x}) \cdot \nabla_\theta f_\theta^j(\mathbf{x}')$, and where the matrix $C_\theta(\mathbf{x}) = K_\theta(\mathbf{x}, \mathbf{x})$ is the previously defined self-correlation matrix $C$. Its role is equivalent of the normalization by $\|\nabla_\theta f_\theta(\mathbf{x})\|^2$ in the 1D case, in plus of decorrelating the gradients.

The interpretation of (1) is that if one moves the output for point $\mathbf{x}$ by $\mathbf{v}$, then the output for point $\mathbf{x}'$ will be moved also, by $M\mathbf{v}$, with $M = K_\theta(\mathbf{x}, \mathbf{x}')\, K_\theta(\mathbf{x}, \mathbf{x})^{-1}$. Note that these matrices $M$ or $K$ are only $d \times d$ where $d$ is the output dimension. They are thus generally small and easy to manipulate or inverse.

## 3.2 Normalized cross-correlation matrix

The normalized version of the kernel (1) is:

$$K_\theta^C(\mathbf{x}, \mathbf{x}') \; = \; C_\theta(\mathbf{x})^{-1/2}\, K_\theta(\mathbf{x}, \mathbf{x}')\, C_\theta(\mathbf{x}')^{-1/2} \tag{2}$$

which is symmetric in the sense that $K_\theta^C(\mathbf{x}', \mathbf{x}) = K_\theta^C(\mathbf{x}, \mathbf{x}')^T$.

A matrix $K_\theta^C(\mathbf{x}, \mathbf{x}')$ with small coefficients means that $\mathbf{x}$ and $\mathbf{x}'$ are relatively independent, from a neural network point of view (moves at $\mathbf{x}$ won't be transferred to $\mathbf{x}'$). On the opposite, the highest possible dependency is $K_\theta^C(\mathbf{x}, \mathbf{x}) = \mathrm{Id}$.

To study properties of this similarity measure, note that $K_\theta^C(\mathbf{x}, \mathbf{x}') = (G_\mathbf{x}^N)^T\, G_{\mathbf{x}'}^N$ with $G_\mathbf{x}^N = G_\mathbf{x}(G_\mathbf{x}^T G_\mathbf{x})^{-1/2}$, where $G_\mathbf{x} = \nabla_\theta f(\mathbf{x})$ : it is the product of normalized, decorrelated versions of the gradient. Indeed, at any point $\mathbf{x}$, the normalized gradient matrix $G_\mathbf{x}^N$ satisfies: $(G_\mathbf{x}^N)^T\, G_\mathbf{x}^N =$

$K_\theta^C(\mathbf{x}, \mathbf{x}) = K_\theta(\mathbf{x}, \mathbf{x})^{-1/2} K_\theta(\mathbf{x}, \mathbf{x}) K_\theta(\mathbf{x}, \mathbf{x})^{-1/2} = \mathrm{Id}$ and consequently $G_\mathbf{x}^N$ can be seen as an orthonormal family of vectors $G_\mathbf{x}^{N,i}$.

The $L^2$ (Frobenius) norm of the ortho-normalized gradient $G_\mathbf{x}^N$ is thus:

$$\left\| G_\mathbf{x}^N \right\|_F^2 = \mathrm{Tr}((G_\mathbf{x}^N)^T G_\mathbf{x}^N) = \mathrm{Tr}(\mathrm{Id}) = d \ .$$

At point $\mathbf{x}'$, $G_{\mathbf{x}'}^N$ is also an orthonormal family, but possibly arranged differently or generating a different subspace of $\mathbb{R}^p$. If $G_\mathbf{x}^N$ and $G_{\mathbf{x}'}^N$ generate the same subspace, then their product $(G_\mathbf{x}^N)^T G_{\mathbf{x}'}^N$ is an orthogonal matrix $Q$ (change of basis) and its $L^2$ (Frobenius) norm is then $\|Q\|_F^2 = \mathrm{Tr}(Q^T Q) = \mathrm{Tr}(\mathrm{Id}) = d$. Otherwise, $(G_\mathbf{x}^N)^T G_{\mathbf{x}'}^N$ can be seen as a projection from one subspace to another one, each vector $G_{\mathbf{x}'}^{N,j}$ is projected onto the ortho-normal family $(G_\mathbf{x}^{N,i})_i$, and as a projection decreases the Euclidean norm, $\sum_i \left( G_\mathbf{x}^{N,i} \cdot G_{\mathbf{x}'}^{N,j} \right)^2 \leqslant \left\| G_{\mathbf{x}'}^{N,j} \right\|^2 = 1$. Thus:

$$\left\| K_\theta^C(\mathbf{x}, \mathbf{x}') \right\|_F = \sqrt{\sum_{ij} \left( G_\mathbf{x}^{N,i} \cdot G_{\mathbf{x}'}^{N,j} \right)^2} \leqslant \sqrt{d} \ .$$

Moreover, any coefficient of the kernel matrix satisfies:

$$\left| K_\theta^{C,ij}(\mathbf{x}, \mathbf{x}') \right| = \left| G_\mathbf{x}^{N,i} \cdot G_{\mathbf{x}'}^{N,j} \right| \leqslant \left\| G_\mathbf{x}^{N,i} \right\|_2 \left\| G_{\mathbf{x}'}^{N,j} \right\|_2 = 1$$

as each vector $G_\mathbf{x}^{N,i}$ is unit-norm. This implies in particular that the trace is bounded:

$$-d \ \leqslant \ \mathrm{Tr}(K_\theta^C(\mathbf{x}, \mathbf{x}')) \ \leqslant d.$$

To sum up, the similarity matrix $K_\theta^C(\mathbf{x}, \mathbf{x}')$ satisfies the following properties:

- its coefficients are bounded, in $[-1, 1]$
- its trace is at most $d$
- its (Frobenius) norm is at most $\sqrt{d}$
- self-similarity is identity: $\forall \mathbf{x}, \ K_\theta^C(\mathbf{x}, \mathbf{x}) = \mathrm{Id}$
- the kernel is symmetric, in the sense that $K_\theta^C(\mathbf{x}', \mathbf{x}) = K_\theta^C(\mathbf{x}, \mathbf{x}')^T$.

### 3.3 Similarity in a single value

Note that when the trace is close to its maximal value $d$, the diagonal coefficients are close to 1, and their contribution to the Frobenius norm squared is close to $d$. Therefore, all non-diagonal coefficients are close to 0, and the matrix is close to $\mathrm{Id}$. And reciprocally, a matrix close to $\mathrm{Id}$ has a trace close to $d$. Thus, two related ways to quantify similarity in a single real value in $[-1, 1]$ appear:

- the distance to the identity $D = \left\| K_\theta^C(\mathbf{x}, \mathbf{x}') - \mathrm{Id} \right\|_F$, which can be turned into a similarity as $1 - \frac{1}{\sqrt{d}} D$ or $1 - \frac{1}{2d} D^2$, since $D \in [0, 2\sqrt{d}]$
- the normalized trace: $\frac{1}{d} \mathrm{Tr}\, K_\theta^C(\mathbf{x}, \mathbf{x}')$, which is also the alignment with the identity: $\frac{1}{d} K_\theta^C(\mathbf{x}, \mathbf{x}') \cdot_F \mathrm{Id}$, where $\cdot_F$ denotes the Frobenius inner product (*i.e.* coefficient by coefficient).

The link between these two quantities can be made explicit by developing:

$$\left\| K_\theta^C(\mathbf{x}, \mathbf{x}') - \mathrm{Id} \right\|_F^2 = \left\| K_\theta^C(\mathbf{x}, \mathbf{x}') \right\|_F^2 - 2\mathrm{Tr}(K_\theta^C(\mathbf{x}, \mathbf{x}')) + d$$

which rewrites as:

$$\left( 1 - \frac{D^2}{2d} \right) = \frac{\mathrm{Tr}(K_\theta^C(\mathbf{x}, \mathbf{x}'))}{d} + \frac{1}{2} \left( 1 - \frac{\left\| K_\theta^C(\mathbf{x}, \mathbf{x}') \right\|_F^2}{d} \right) .$$

The last term lies in $[0, 1]$ and measures the mismatch between the vector subspaces generated by the two families of gradients $\left( \nabla_\theta f^i(\mathbf{x}) \right)_i$ and $\left( \nabla_\theta f^i(\mathbf{x}') \right)_i$. It is 1 when $f_\theta(\mathbf{x})$ and $f_\theta(\mathbf{x}')$ can be moved independently, and 0 when they move jointly (though not necessarily in the same direction).

As our two similarity measures $1 - \frac{D^2}{2d}$ and $\frac{1}{d} \mathrm{Tr}(K_\theta^C(\mathbf{x}, \mathbf{x}'))$ have same optimum ($\mathrm{Id}$) and are closely related, in the sequel we will focus on the second one and define:

$$k_\theta^C(\mathbf{x}, \mathbf{x}') \ = \ \frac{1}{d} \mathrm{Tr}\, K_\theta^C(\mathbf{x}, \mathbf{x}') \ . \tag{3}$$

## 3.4 Metrics on output: rotation-invariance

Similarity in $\mathbb{R}^d$, to compare $\mathbf{v}$ and $\mathbf{v}' = M\mathbf{v}$, might be richer than just checking whether the vectors are equal or close in $L^2$ norm.

For instance, one could quotient the output space by the group of rotations, in order to express a known or desired equivariance of the network to rotations. If the output is the predicted motion of some object described in the input, one could wish indeed that if the input object is rotated by an angle $\phi$, then the output should be rotated as well with the same angle.

In that case, given two inputs $\mathbf{x}$ and $\mathbf{x}'$ and associated output variations $\mathbf{v}$ and $\mathbf{v}'$, without knowing the rotation angle if applicable, one could consider all possible rotated versions $R_\phi \mathbf{v}' = R_\phi M \mathbf{v}$, where $R_\phi$ is the rotation matrix with angle $\phi$, and pick the best angle $\phi$ that maximizes the alignment $\mathbf{v} \cdot R_\phi M \mathbf{v}$, *i.e.* such that $R_\phi M$ is the closest to the $d \times d$ identity matrix. This can be computed easily in closed form, for instance in the 2-dimensional case as follows.

The $2 \times 2$ matrix of interest (Eq. 2) can be written as the product of two $p \times 2$ matrices of the form $G(G^T G)^{-1/2}$, where $G$ is the matrix containing the gradient of all coordinates. Rotating the coordinates of $G$ amounts to considering $G R_\phi (R_\phi^T G^T G R_\phi)^{-1/2} = G(G^T G)^{-1/2} R_\phi$ instead. Thus the effect of rotation is just right-multiplying our $2 \times 2$ matrix $M$ of interest (Eq. 2) by $R_\phi$. We are thus interested into getting $M R_\phi$ as close as possible to the $2 \times 2$ identity. For our trace-based similarity kernel (Eq. 3), this amounts to maximizing $\mathrm{Tr}(M R_\phi) = \cos(\phi)(M_{11} + M_{22}) + \sin(\phi)(M_{12} - M_{21})$ w.r.t. $\phi$, whose optimal value is:

$$
\begin{aligned}
k_\theta^{C,\mathrm{rot}}(\mathbf{x}, \mathbf{x}') &= \frac{1}{2}\sqrt{(M_{11} + M_{22})^2 + (M_{12} - M_{21})^2} \\
&= \frac{1}{2}\sqrt{\|M\|_F^2 + 2\det M}
\end{aligned}
$$

where $M = K_\theta^C(\mathbf{x}, \mathbf{x}')$. This quantity is indeed rotation-invariant, as the Frobenius norm and the determinant do not change upon rotations. Note that one could also consider instead the subspace match $\frac{1}{d}\|M\|_F^2$. The main difference between the two is that the first one penalizes mirror symmetries (through $\det M$) while the second one does not.

Note that other metrics are possible in the output space. For instance, the loss metric quantifies the norm of a move $\mathbf{v}$ by its impact on the loss $\left.\frac{dL(y)}{dy}\right|_{f_\theta(\mathbf{x})}(\mathbf{v})$. It has a particular meaning though, and is relevant only if well designed and not noisy, as seen in the remote sensing image registration example.

# 4 Estimating density

## 4.1 Toy problem

The toy problem used in the paper to test the various estimators for neighbor count estimation consists of predicting a one dimensional function, namely a sinusoid (such as in Fig.1 (a)). We can easily change the difficulty of the problem by using different values of frequency. The neural network would perform this mapping: $y = \sin(2\pi f x), x \in [0, 1]$.

A problem arises however when estimating the number of neighbors because the input space has 2 boundaries at $x = 0$ and $x = 1$, leading to fewer neighbors when $x$ approaches either of those boundaries. To avoid this problem, we transform the input space to a 2D circle. Namely, the task is now $y = sin(2\pi f \alpha(x)), x \in \{(\cos(2\pi\alpha), \sin(2\pi\alpha)), \alpha \in [0, 1]\}$, with the input space having no boundaries.

The dataset is generated with n=2048 input points. The network used is fully-connected and has 5 hidden layers of 64 neurons trained with the Adam optimizer for 80 epochs with a base learning rate of $1e^{-4}$. An experiment consist of training the network on a dataset generated with a specific frequency f. Each experiment was repeated 5 times, in order to take the median of every result to limit the variance due to the neural network stochastic training.

We can see in Fig.1 (b) the proposed soft estimate $k_\theta^C$ for each input point (projected to 1D). As expected we observe that the number of neighbors drops when the curvature is high: the objective changes quickly and the network adjusts to better distinguish inputs in places of higher curvature.

## 4.2 Other possible uses

**Density homogeneity as an optimization criterion**  The estimations above are meant to be done post-training. This said, one could control density explicitly, by computing the number of neighbors for all points, and asking it to be in a reasonable range, or in a reasonable proportion $q$ of the dataset size $\mathcal{D}$, by adding *e.g.* to the loss $\sum_i \left( \frac{N_S(\mathbf{x}_i)}{\mathcal{D}} - q \right)^2$. Online learning could also make use of such tools, to sample first lowly-populated areas, where uncertainty is higher.

## 5 Enforcing similarity

The similarity criterion we defined could be used not only to estimate how similar two samples are perceived, after training, but also to incite the network, during training, to evolve in order to consider these samples as similar.

**Asking two samples to be treated as similar**  If two inputs $\mathbf{x}$ and $\mathbf{x}'$ are known to be similar (from a human point of view), one can enforce their similarity from the network perspective, by adding to the loss the term:
$$-k_\theta^C(\mathbf{x}, \mathbf{x}') .$$

**Asking a distribution of samples to be treated as similar**  By extension, to enforce the similarity of a subset $\mathcal{S}$ of training samples, of size $n = |\mathcal{S}|$, one might consider the average pairwise similarity $k_\theta^C$ over all pairs, or the standard deviation of the gradients. Both turn out to be equivalent to maximizing the norm of the gradient mean $\mu = \frac{1}{n} \sum_{i \in \mathcal{S}} \frac{\nabla_\theta f_\theta(\mathbf{x}_i)}{\|\nabla_\theta f_\theta(\mathbf{x}_i)\|}$:

$$\frac{1}{n(n-1)} \sum_{i,j \in \mathcal{S}, i \neq j} k_\theta^C(x_i, x_j) = \frac{n}{n-1}\|\mu\|^2 - \frac{1}{n-1} \quad \text{and} \quad \operatorname*{var}_{i \in \mathcal{S}} \frac{\nabla_\theta f_\theta(\mathbf{x}_i)}{\|\nabla_\theta f_\theta(\mathbf{x}_i)\|} = 1 - \|\mu\|^2 .$$

In practice, common deep learning platforms are much faster when using mini-batches, but then return only the gradient sum $\sum_{i \in \mathcal{B}} \nabla_\theta f_\theta(\mathbf{x}_i)$ over a mini-batch $\mathcal{B}$, not individual gradients, preventing the normalization of each of them to compute $k_\theta^C$ or $\mu$. So instead we compare means of un-normalized gradients, over two mini-batches $\mathcal{B}_1$ and $\mathcal{B}_2$ comprising each $n_B$ samples from $\mathcal{S}$, which yields the criterion:

$$n_B \frac{\|\mu_1 - \mu_2\|^2}{\|\mu_1\|\|\mu_2\|} \quad \text{where} \quad \mu_k = \frac{1}{n} \sum_{i \in \mathcal{B}_k} \nabla_\theta f_\theta(\mathbf{x}_i) .$$

The factor $n_B$ counterbalances the $\frac{1}{\sqrt{n_B}}$ variance reduction effect due to averaging over $n_B$ samples.

### 5.1 Group invariance

Dataset augmentation is a standard machine learning technique; when augmenting the dataset by a group transformation of the input (*e.g.*, translation, rotation...) or by small intensity noise, new samples are artificially created, to augment the dataset size and hope for invariance to such transformations. One can ask the network to consider orbits of samples [2] as similar with the technique above.

Furthermore, if the group infinitesimal elements are expressible as differential operators $e_k$, one could require directly, for all $\mathbf{x}$, invariance in the tangent plane in the directions of these differential operators:
$$\|\partial_\mathbf{x} \nabla_\theta f(\mathbf{x}) \cdot e_k(\mathbf{x})\|^2$$
which is the limit of $\frac{1}{\varepsilon^2}\|\nabla_\theta f_\theta(\mathbf{x}) - \nabla_\theta f_\theta(\mathbf{x} + \varepsilon e_k(\mathbf{x}))\|^2$ when $\varepsilon \to 0$. For instance, in the case of image translations, the operator is $e : \mathbf{x} \mapsto \nabla_x \mathbf{x}(x)$ where $x$ denotes spatial coordinates in the image $\mathbf{x}$, as $\mathbf{x}(x + \tau) = \mathbf{x}(x) + \tau \cdot \nabla_x \mathbf{x}(x) + O(\tau^2)$. This is however not recommended, as representing a translation with such a spatially-local operator does not take into account the spatially-irregular nature of image intensities.

(a) Function to predict.

(b) Neighbors soft estimate.

Figure 1: Toy problem with the frequency f = 2.

Figure 2: 3D plot of neighbors soft with varying frequency. Script and data to plot interactively in attached files. Run the bash script "main_plot_exps.paper.sh" to reproduce this exact figure. Alternatively use "main_plot_exps.py" with arguments of your choosing to plot different values (run "python main_plot_exps.py -h" to see possible arguments).

Note that to the opposite of standard robustification techniques considering regularizers such as $\sum_\mathbf{x} \|\nabla_\theta f_\theta(\mathbf{x})\|^2$, we ask not gradients to be always small, but to be smooth, and in certain directions only.

**Complexity**    A gradient descent step on our criterion for a given pair $(\mathbf{x}, \mathbf{x}')$ (in a mini-batch approach, *e.g.*) requires the computation of the gradient $\nabla_\theta k_\theta^C(\mathbf{x}, \mathbf{x}') = \nabla_\theta \left( \nabla_\theta f_\theta(\mathbf{x}) \cdot \nabla_\theta f_\theta(\mathbf{x}') \right)$. While a naive approach would require the computation of a second derivative, *i.e.* a matrix of size $p \times p$ where $p$ is the number of parameters, it is actually possible to compute $\nabla_\theta k_\theta^C(\mathbf{x}, \mathbf{x}') = \nabla_\theta \sum_i \frac{df_\theta(\mathbf{x})}{d\theta_i} \frac{df_\theta(\mathbf{x}')}{d\theta_i}$ in linear time $O(p)$, taking advantage of the serial structure of the computational graph. The framework enabling such computations is already available on common deep learning platforms, under the name of *double-backpropagation* routine [3, 6, 10, 5], roughly doubling the computational time of a gradient step. It was initially intended for the computation of $\nabla_\mathbf{x} \nabla_\theta f_\theta(\mathbf{x})$ for some variations on GANs.

**Dynamics of learning**    Our approach enforces similarity not just at the output level, but within the whole internal computational process. Therefore, during training, information is provided directly to each parameter instead of being back-propagated through possibly many layers. Thus the dynamics of learning are expected to be different, especially for deep networks.

To test this hypothesis, we train a small network on MNIST with and without the similarity criteria acting as an auxiliary loss (see Fig. 3). As a result, we observe an acceleration of the convergence very early in the learning process. It is worth noting that this effect can be observed across a wide range of different neural architectures. We performed additional experiments on toy datasets as well as on CIFAR10 with no or only negligible improvements. All together this suggests that using the similarity criteria during training may be beneficial to specific datasets as opposed to specific architectures, and indeed, as the class intra-variability in CIFAR10 is known to be high, considering all examples of a class of CIFAR10 as similar is less relevant.

Figure 3: Validation accuracy of a neural network trained on MNIST with and without the similarity criterion (note that the x-axis is the number of minibatches presented to the network, not of epochs).

**Experimentation details**    The results in Figure 3 show the average and standard deviation over 60 runs for each curve. The x-axis is the number of batches to the network is trained on (with a batch size of 16). The y-axis is the accuracy metric on the whole validation set. The network architecture is made of 2 convolutions layers (with a kernel size of 5), 2 linear layers and uses PReLU non-linearities. We used Adam with a learning rate of 1e-3 and no weight decay.

We tested other architectures on MNIST: one with residual blocks, one deeper (8 convolutions) and one with tanh non-linearities. Similar results were observed on all cases. Additional tests were performed on CIFAR10 with a VGG architecture and only negligible benefits were observed.

# 6  Noisy Map Alignment Analysis

The task here it to align maps in the form of a list of polygons with remote sensing images while using only the available noisy annotations. We analyze the model developed in a previous work [4]. Specifically, the model is trained in a multiple-rounds training scheme to iteratively align the available noisy annotations, which provides a better ground truth used to train a better model in the next round. We aim to answer the question of why multiple rounds are needed in this noisy supervision setting, and why not all the noise can be removed in a single training step.

More specifically, the model is made out of 4 neural networks. Each is trained on a different resolution (in terms of ground pixel size) and are applied in a multi-resolution pyramidal manner. In all our experiments we only analyzed the networks trained for a ground pixel size of 4 time smaller than the reference ground pixel size which is $0.3m$. We used the already-trained networks for each round, of which there are 3.

The network was trained with small patches of (image, misaligned map) pairs from images of the Inria dataset [8] and the Bradbury dataset [1]. Ideally we would want to compute the similarities of every possible pairs of inputs, with a small patch size of $124$ px. However, given that a typical image of the training dataset is $1250 \times 1250$ px (after rescaling) and there are a few hundred of them (328 from the Inria dataset, only counting images where OSM annotations [9]), this would result in 32800 patches. The resulting amount of similarities to compute would be around half a billion. As the network has a few million of parameters and the output is 2D, each computation of similarity takes around 0.5s. To make any computation feasible, we first sample 10 patches per image from the 328 of the Inria dataset. Those patches are chosen at random, as long as there is at least one building lying fully in the patch. As some images have rather sparse buildings, some images give less than 10 patches. We thus obtain 3045 patches representing the dataset. The amount of similarities to compute would be close to 5 million. To study all patches globally, we can use the soft neighbors estimator $k_\theta^C$ which has a linear complexity and allows us to compute the amount of neighbors for all 3045 patches in under an hour on a GTX 1080Ti. However it is also interesting to go in deeper detail and compute similarities for some input pairs. We thus furthermore reduce the amount of pairs by estimating all similarities only for a very small number of patches, for example 10. This results in a $10 \times 3045$ similarity matrix.

## 6.1  Soft estimate on a sampling of the training dataset

In this section we present the results of computing the soft neighbors estimator $k_\theta^C$ on the 3045 sampled patches of inputs. We obtain results for the 3 networks of the 3 rounds of the noisy-supervision multi-rounds training scheme. Fig.4 shows a histogram of the soft neighbors estimations. It additionally representative input patches for each bin of the histogram. Those representative patches are chosen so that their neighbor count is closest to the right edge of that bin. We especially observe that inputs in round 2 have more neighbors than the other 2 rounds. This particularity of round 2 will be seen throughout the remaining results. It is the round that aligns the most the annotations (see the Fig.2 on accuracy cumulative distributions in the paper). Round 3 does not perform any more alignment, that might be the reason why its results are different from those of round 2.

## 6.2  Similarities on pairs of input patches

In this section are the results for the computation of similarities between pairs of input patches. In a first experiment, for every round we chose the 10 patches shown in Fig.4, and computed their similarities with all the other 3045 patches. In order to visualize this data, we computed the 10-nearest neighbors in terms of similarity for each of those patches, see Fig.5, 6, 7. We computed the histogram of similarities as well, see Fig.8.

In a second experiment, to better compare between rounds, we used another set of 10 patches, this time the same set for each round. Specifically, we sampled 10 patches from the bloomington22 image of the Inria dataset. As just before we computed the 10-nearest neighbors (Fig.9, 10, 11) and the histogram of similarities(Fig.12) for a visualization of those measures.

Generally speaking, inputs in round 2 have more neighbors and the 10-nearest ones are closer than in other rounds (see Fig.5, 6, 7 and Fig.9, 10, 11). For each parch, its closest neighbors generally (for similarity > 0.8) look similar from a human point of view. For example patches with sparse houses

(a) Round 1

(b) Round 2

(c) Round 3

Figure 4: Histogram of the soft estimate of neighbors on 3045 patches. Horizontal scale is different for each.

Figure 5: **Round 1**: k-nearest neighbors with k=10. The 10 patches selected correspond to the 10 patches of Fig.4 for that round.

and trees have the same kind of neighbors. The same can be said for patches with parking lots and big roads. Another group are patches that are almost empty of buildings, with a lot of low vegetation. Other patch nearest neighbors are more difficult to interpret. In Fig.8 and Fig.12 we can see that for round 2, the spread of the similarities of the selected patches is smaller and the peak of the histogram are closer to the right, meaning all patches are closer than in other rounds. Additionally in Fig.8 we can observe that the bottom patch has closer neighbors than the top patch, this is because the top patch corresponds to the left patch in 4 and the bottom one corresponds to the right patch in 4.

We additionally computed a perceptual loss on the same 10 patches from the bloomington22 image for comparison (see Fig.13, 14, 15). We used the activations from the second to last layer of the displacement branch of our network to compute it. In general it does not estimate neighbors as well, especially regarding the type of buildings and their arrangement in the image patch.

Figure 6: **Round 2**: k-nearest neighbors with k=10. The 10 patches selected correspond to the 10 patches of Fig.4 for that round.

Figure 7: **Round 3**: k-nearest neighbors with k=10. The 10 patches selected correspond to the 10 patches of Fig.4 for that round.

Figure 8: Histograms of similarities shown for the same 10 patches as Fig.4 and Fig.5, 6, 7.

Figure 9: **Round 1**: k-nearest neighbors with k=10. The 10 patches are from from the bloomington22 image. Same patch selection across rounds.

Figure 10: **Round 2**: k-nearest neighbors with k=10. The 10 patches are from from the bloomington22 image. Same patch selection across rounds.

Figure 11: **Round 3**: k-nearest neighbors with k=10. The 10 patches are from from the bloomington22 image. Same patch selection across rounds.

Figure 12: Histograms of similarities shown for the same 10 patches as in Fig.9, 10, 11. Same patch selection across rounds.

Figure 13: **Round 1**: perceptual loss used for k-nearest neighbors with k=10. The 10 patches are from from the bloomington22 image. Same patch selection across rounds.

Figure 14: **Round 2**: perceptual loss used for k-nearest neighbors with k=10. The 10 patches are from from the bloomington22 image. Same patch selection across rounds.

Figure 15: **Round 3**: perceptual loss used for k-nearest neighbors with k=10. The 10 patches are from from the bloomington22 image. Same patch selection across rounds.

## 6.3 Proof details of the self-denoising effect quantification

### 6.3.1 Magnitude of kernel-smoothed i.i.d. noise

We show here that $\mathbb{E}_k[\varepsilon] \propto \mathrm{var}_\varepsilon(\mathbb{E}_k[\varepsilon])^{1/2} = \sigma_\varepsilon \|k_\theta^{IN}\|_{L2}$.

Let us denote by $\mathbb{E}_\varepsilon[\,]$ and $\mathrm{var}_\varepsilon(\,)$ the expectation and variance with respect to the random variable $\varepsilon$. As a reminder, by assumptions in the noise definition, $\varepsilon = (\varepsilon_i)_i$ is a random, i.i.d. noise, centered and of variance $\sigma_\varepsilon$.

This is not to be confused with the symbol $\mathbb{E}_k[\,]$, which was defined as, for any vector field $a$:

$$\mathbb{E}_k[a] = \sum_j a_j \, k_\theta^{IN}(\mathbf{x}_j, \mathbf{x}_i) \,,$$

*i.e.* as the mean value of $a$ in the neighborhood of $i$, that is, the weighted average of the $a_j$ with weights $k_\theta^{IN}(\mathbf{x}_j, \mathbf{x}_i)$, which are positive and sum up to 1.

Given a network and its associated kernel $k_\theta^{IN}$, we are interested in to knowing the typical values of $\mathbb{E}_k[\varepsilon]$ for random $\varepsilon$. First, the expectation over the noise of $\mathbb{E}_k[\varepsilon]$ is:

$$\mathbb{E}_\varepsilon\left[\mathbb{E}_k[\varepsilon]\right] \;=\; \mathbb{E}_\varepsilon\left[\sum_j \varepsilon_j \, k_\theta^{IN}(\mathbf{x}_j, \mathbf{x}_i)\right] \;=\; \sum_j \mathbb{E}_\varepsilon[\varepsilon_j]\, k_\theta^{IN}(\mathbf{x}_j, \mathbf{x}_i) \;=\; 0$$

as $\varepsilon$ is a centered noise. Thus the random variable $\mathbb{E}_k[\varepsilon]$ is also centered, and therefore its typical values are described by its standard deviation, which is the square root of its variance:

$$\mathbb{E}_k[\varepsilon] \;\propto\; \mathrm{var}_\varepsilon\left(\mathbb{E}_k[\varepsilon]\right)^{1/2} \,.$$

The variance can be computed as follows:

$$
\begin{aligned}
\mathrm{var}_\varepsilon\left(\mathbb{E}_k[\varepsilon]\right) &= \mathbb{E}_\varepsilon\left[\left(\sum_j \varepsilon_j \, k_\theta^{IN}(\mathbf{x}_j, \mathbf{x}_i)\right)^2\right] \\
&= \mathbb{E}_\varepsilon\left[\sum_j \varepsilon_j^2 \left(k_\theta^{IN}(\mathbf{x}_j, \mathbf{x}_i)\right)^2\right] \quad \text{as } \varepsilon \text{ is i.i.d.} \\
&= \sigma_\varepsilon^2 \sum_j \left(k_\theta^{IN}(\mathbf{x}_j, \mathbf{x}_i)\right)^2 \\
&= \sigma_\varepsilon^2 \left\|k_\theta^{IN}(\cdot, \mathbf{x}_i)\right\|_{L2}^2 \,.
\end{aligned}
$$

As the weights $p_j = k_\theta^{IN}(\mathbf{x}_j, \mathbf{x}_i)$, for given $i$ and varying $j$, are positive and sum up to 1, they form a probability distribution. Hence the value of $\left\|k_\theta^{IN}(\cdot, \mathbf{x}_i)\right\|_{L2}^2 = \|p\|_{L2}^2$ satisfies:

- $\|p\|_{L2} \leqslant 1$, as $\sum_j p_j^2 \leqslant \sum_j p_j = 1$, with equality only when $p_j = p_j^2 \; \forall j$, that is, all $p_j = 0$ except for one $p_{j*} = 1$, which means $k_\theta^I(\mathbf{x}_j, \mathbf{x}_i) = 0 \quad \forall j \neq i$, which means that all data samples are fully independent from the network's point of view.

- $\|p\|_{L2} \geqslant \frac{1}{\sqrt{N}}$ as $1 = \sum_j 1 \times p_j \leqslant \|1\|_{L2}\, \|p\|_{L2} = \sqrt{N}\, \|p\|_{L2}$ (Cauchy-Bunyakovsky-Schwarz), with equality reached for the uniform distribution: $p_j = \frac{1}{N} \; \forall j$, where $N$ is the number of data samples. This implies that all $k_\theta^C(\mathbf{x}_j, \mathbf{x}_i)$ are equal, for all $i, j$, hence they are all equal to $k_\theta^C(\mathbf{x}_i, \mathbf{x}_i) = 1$. This is the case studied in [7]: all input points are identical.

The denoising factor $\|k_\theta^{IN}(\cdot, \mathbf{x}_i)\|_{L2}$, which depends on the data point $\mathbf{x}_i$ considered, thus expresses where the neighborhood of $\mathbf{x}_i$ lies, between these two extremes (all $\mathbf{x}_j$ very different from $\mathbf{x}_i$, or all identical).

Note: the results above remain valid when the output is higher-dimensional, under the supplementary assumption that the covariance matrix of the noise is proportional to the Identity matrix (*i.e.*, the noises on the various coefficients of the label vector are independent from each other, and follow the same law, with standard deviation $\sigma_\varepsilon$). If not, the expression for $\mathrm{covar}_\varepsilon\left(\mathbb{E}_k[\varepsilon]\right)$ is more complex, as $\Sigma_\varepsilon$ and $k_\theta^{IN}$ interact. Note that when the output is of dimension $d$, the kernel $k_\theta^{IN}(\mathbf{x}_j, \mathbf{x}_i)$ is a $d \times d$ matrix, thus the denoising factor $\left\| k_\theta^{IN}(\cdot, \mathbf{x}_i) \right\|_{L2}^2$ has to be replaced with the matrix $\sum_j k_\theta^{IN}(\mathbf{x}_j, \mathbf{x}_i)\, k_\theta^{IN}(\mathbf{x}_j, \mathbf{x}_i)^T$, which can be summarized by its trace, which is the $L^2$ norm of the Frobenius norms: $\left\| \left\| k_\theta^{IN}(\cdot, \mathbf{x}_i) \right\|_F \right\|_{L2}^2$.

### 6.3.2 The function: gradient $\mapsto$ output is Lipschitz

Theorem 1 implies that the application: $\frac{\nabla_\theta f_\theta(\mathbf{x})}{\|\nabla_\theta f_\theta(\mathbf{x})\|} \mapsto f_\theta(\mathbf{x})$ is well-defined. We show here that this application is also Lipschitz, with a network-dependent constant, under mild hypotheses.

We consider the same assumptions as in Theorem 1 : $f_\theta$ is a real-valued network, whose last layer is a linear layer or a standard activation function thereof (such as sigmoid, tanh, ReLU...), without parameter sharing (in that last layer). We will also require that the derivative of the activation function is bounded, which is a safe assumption for all networks meant to be trained by gradient descent. Another, technical property (bounded input space) will be assumed in order to imply bounded gradients. A side note indicates how to rewrite the desired property if the input space is not bounded.

Let $\mathbf{x}$ and $\mathbf{x}'$ be any two inputs. We want to bound $|f_\theta(\mathbf{x}) - f_\theta(\mathbf{x}')|$ by $\|\mathbf{u} - \mathbf{u}'\|_2$ times some constant, where $\mathbf{u} = \frac{\nabla_\theta f_\theta(\mathbf{x})}{\|\nabla_\theta f_\theta(\mathbf{x})\|}$ and $\mathbf{u}' = \frac{\nabla_\theta f_\theta(\mathbf{x}')}{\|\nabla_\theta f_\theta(\mathbf{x}')\|}$.

Let us denote the non-normalized gradients by $\mathbf{v} = \nabla_\theta f_\theta(\mathbf{x})$ and $\mathbf{v}' = \nabla_\theta f_\theta(\mathbf{x}')$. We have $\mathbf{u} = \frac{\mathbf{v}}{\|\mathbf{v}\|}$ and $\mathbf{u}' = \frac{\mathbf{v}'}{\|\mathbf{v}'\|}$.

We will proceed in two steps: bounding $|f_\theta(\mathbf{x}) - f_\theta(\mathbf{x}')|$ by $\|\mathbf{v} - \mathbf{v}'\|_2$, and then $\|\mathbf{v} - \mathbf{v}'\|_2$ by $\|\mathbf{u} - \mathbf{u}'\|_2$. The first step is easy and actually sufficient to bound with a non-normalized similarity kernel $k_\theta = \mathbf{v} \cdot \mathbf{v}'$ the shift from the average prediction in the neighborhood. The second step provides a more elegant bound, in that it makes use of the normalized similarity kernel $k_\theta^C = \mathbf{u} \cdot \mathbf{u}'$, but that bound is a priori not as tight and requires more assumptions.

**Case where the last layer is linear**

The output of the network is of the form

$$f_\theta(\mathbf{x}) = \sum_i w_i a_i(\mathbf{x}) + b \, ,$$

where $w_i$ and $b$ are parameters in $\mathbb{R}$ and $a_i(\mathbf{x})$ activities from previous layers. Thus:

$$
\begin{aligned}
|f_\theta(\mathbf{x}) - f_\theta(\mathbf{x}')| &= \left| \sum_i w_i \left( a_i(\mathbf{x}) - a_i(\mathbf{x}') \right) \right| \\
&\leqslant \|\mathbf{w}\|_2 \, \|\mathbf{a}(\mathbf{x}) - \mathbf{a}(\mathbf{x}')\|_2 \\
&\leqslant \|\mathbf{w}\|_2 \sqrt{\sum_i (v_i - v_i')^2}
\end{aligned}
$$

where the sum is taken over parameters $i$ in the last layer only, using the fact that activities $a_i$ in the last layer are equal to some of the coefficients of the gradient: $v_i := \frac{\partial f_\theta(\mathbf{x})}{\partial w_i} = a_i(\mathbf{x})$.

Note that the derivative with respect to the shift $b$ is $v_b := \frac{\partial f_\theta(\mathbf{x})}{\partial b} = 1$, which ensures that the norm of $\mathbf{v}$ is at least 1. This implies:

$$\|\mathbf{u} - \mathbf{u}'\|_2 \geqslant |u_b - u_b'| = \left| \frac{1}{\|\mathbf{v}\|} - \frac{1}{\|\mathbf{v}'\|} \right|$$

which, combined with:

$$|v_i - v_i'| = \|\mathbf{v}'\| \left| \frac{1}{\|\mathbf{v}'\|} v_i - \frac{v_i'}{\|\mathbf{v}'\|} \right| = \|\mathbf{v}'\| \left| u_i - u_i' + \left( \frac{1}{\|\mathbf{v}'\|} - \frac{1}{\|\mathbf{v}\|} \right) v_i \right|$$

yields:

$$|v_i - v_i'| \;\leqslant\; \|\mathbf{v}'\| \left( |u_i - u_i'| + \|\mathbf{u} - \mathbf{u}'\|_2\, |v_i| \right) \;\leqslant\; \|\mathbf{v}'\|\, \|\mathbf{u} - \mathbf{u}'\|_2 \left( 1 + |v_i| \right)$$

from which we finally obtain:

$$|f_\theta(\mathbf{x}) - f_\theta(\mathbf{x}')| \;\leqslant\; \left[ \|\mathbf{w}\|_2\, \|\mathbf{v}'\| \sqrt{\sum_i (1 + |v_i|)^2} \right] \|\mathbf{u} - \mathbf{u}'\|_2$$

which is the bound we were searching for. For the term between brackets to be bounded by a network-dependent constant, one can suppose for instance that the derivative of the activation functions is bounded (which is usually the case for networks meant to be trained by gradient descent), and that the input space is bounded as well; in such cases indeed all coefficients of the gradient vector $\mathbf{v}$ or $\mathbf{v}'$ are bounded, as derivatives of a function composed of constant linear applications (except for the first layer which is a linear application whose factors are bounded inputs, when seen as an application defined on parameters) and of bounded-derivatives activation functions.

**Note for unbounded input spaces:** If the input space is not bounded, the gradients are not bounded absolutely, as for instance the gradient with respect to a weight in the first layer is the input itself (times a chain product). In that case the application $\mathbf{x} \mapsto \mathbf{v}$ still satisfy a bound of the form $\|\mathbf{v}\| \leqslant (1 + \|\mathbf{x}\|)\, A$, with $A$ a network-dependent constant (product of determiners of layer weight matrices and of the bound on activation function derivatives to the power: network depth), and thus the application $\mathbf{u} \mapsto f_\theta(\mathbf{x})$ still satisfies a bound of the form, for any $\mathbf{x}, \mathbf{x}'$:

$$|f_\theta(\mathbf{x}) - f_\theta(\mathbf{x}')| \;\leqslant\; B\, (1 + \|\mathbf{x}\|)\, (1 + \|\mathbf{x}'\|)\, \|\mathbf{u} - \mathbf{u}'\|_2 \;.$$

The last statement in the paper then becomes

$$\left| \underset{k}{\mathbb{E}}[\widehat{y_i} - \widehat{y}] \right| \;\leqslant\; \sqrt{2}\, B\, (1 + \|\mathbf{x}_i\|)\, \max_j (1 + \|\mathbf{x}_j\|)\, \underset{k}{\mathbb{E}}\left[ \sqrt{1 - k_\theta^C(\mathbf{x}_i, \cdot)} \right]$$

which in practice rewrites as the original formulation:

$$\left| \underset{k}{\mathbb{E}}[\widehat{y_i} - \widehat{y}] \right| \;\leqslant\; \sqrt{2}\, C\, \underset{k}{\mathbb{E}}\left[ \sqrt{1 - k_\theta^C(\mathbf{x}_i, \cdot)} \right]$$

by taking $C = B \max_j (1 + \|\mathbf{x}_j\|)^2$, considering the actual diameter of the given dataset.

**Case where the last layer is an activation function of a linear layer**

The output of the network is of the form

$$f_\theta(\mathbf{x}) = \sigma \left( \sum_i w_i a_i(\mathbf{x}) + b \right) \;,$$

and, as the derivative of $\sigma$ is assumed to be bounded, and as the weights $w_i$ are fixed, $f_\theta(\mathbf{x})$ is a Lipschitz function of the last layer activities $a_i(\mathbf{x})$. Therefore:

$$|f_\theta(\mathbf{x}) - f_\theta(\mathbf{x}')| \leqslant K \|\mathbf{a}(\mathbf{x}) - \mathbf{a}(\mathbf{x}')\|_2 \;.$$

We will denote by $\alpha$ and $\alpha'$ the derivatives with respect to the shift $b$, which are this time:

$$\alpha := \mathbf{v}_b := \frac{\partial f_\theta(\mathbf{x})}{\partial b} = \sigma' \Big|_{\sum_i w_i a_i(\mathbf{x}) + b} \quad \text{and} \quad \alpha' := \mathbf{v}_b' := \frac{\partial f_\theta(\mathbf{x}')}{\partial b} = \sigma' \Big|_{\sum_i w_i a_i(\mathbf{x}') + b} \;.$$

We proceed as previously:

$$\|\mathbf{u} - \mathbf{u}'\|_2 \;\geqslant\; |u_b - u_b'| \;=\; \left| \frac{\alpha}{\|\mathbf{v}\|} - \frac{\alpha'}{\|\mathbf{v}'\|} \right|$$

which, combined with:

$$|a_i - a_i'| \;=\; \left| \frac{v_i}{\alpha} - \frac{v_i'}{\alpha'} \right| \;=\; \frac{\|\mathbf{v}'\|}{\alpha'} \left| \frac{\alpha'}{\alpha \|\mathbf{v}'\|} v_i - \frac{v_i'}{\|\mathbf{v}'\|} \right| \;=\; \frac{\|\mathbf{v}'\|}{\alpha'} \left| u_i - u_i' + \frac{v_i}{\alpha} \left( \frac{\alpha'}{\|\mathbf{v}'\|} - \frac{\alpha}{\|\mathbf{v}\|} \right) \right|$$

yields:

$$|a_i - a_i'| \; \leqslant \; \frac{\|\mathbf{v}'\|}{\alpha'} \left( \, | \, u_i - u_i' \, | \, + \, \|\mathbf{u} - \mathbf{u}'\|_2 \, |a_i| \right) \; \leqslant \; \frac{\|\mathbf{v}'\|}{\alpha'} \left( 1 + |a_i| \right) \|\mathbf{u} - \mathbf{u}'\|_2$$

from which we finally obtain:

$$|f_\theta(\mathbf{x}) - f_\theta(\mathbf{x}')| \; \leqslant \; \left[ K \, \frac{\|\mathbf{v}'\|}{\alpha'} \, \sqrt{\sum_i (1 + |a_i|)^2} \, \right] \, \|\mathbf{u} - \mathbf{u}'\|_2 \; .$$

Note that $\alpha'$ is actually a factor of each coefficient of $\mathbf{v}'$, as the derivative of $f_\theta(\mathbf{x}')$ with respect to any parameter is a chain rule starting with $\frac{\partial f_\theta(\mathbf{x}')}{\partial b} = \sigma'\Big|_{\sum_i w_i a_i(\mathbf{x}') + b} = \alpha'$. To bound the term between brackets, the same assumptions as previously are sufficient. One can assume that $\alpha$ and $\alpha'$ are not 0, as, if they are, the problem is of little interest ($\mathbf{u}$ or $\mathbf{u}'$ being then not defined).

### 6.3.3 Additional proof detail

The kernel $k_\theta^C(\mathbf{x}, \mathbf{x}')$, by definition, is the $L^2$ inner product between two unit vectors:

$$k_\theta^C(\mathbf{x}, \mathbf{x}') = \frac{\nabla_\theta f_\theta(\mathbf{x})}{\|\nabla_\theta f_\theta(\mathbf{x})\|} \cdot \frac{\nabla_\theta f_\theta(\mathbf{x}')}{\|\nabla_\theta f_\theta(\mathbf{x}')\|} \; .$$

As, for any two unit vectors $a$ and $b$:

$$\|a - b\|^2 \; = \; a^2 + b^2 - 2 \, a \cdot b \; = \; 2 \left( 1 - a \cdot b \right),$$

we get:

$$\left\| \frac{\nabla_\theta f_\theta(\mathbf{x})}{\|\nabla_\theta f_\theta(\mathbf{x})\|} - \frac{\nabla_\theta f_\theta(\mathbf{x}')}{\|\nabla_\theta f_\theta(\mathbf{x}')\|} \right\| = \sqrt{2}\sqrt{1 - k_\theta^C(\mathbf{x}, \mathbf{x}')} \; .$$

## 7 Other related remarks

**Data augmentation** can be seen as label denoising, as it multiplies the number of neighbors. Indeed, in the infinite sampling limit, where the dataset becomes a probability distribution over all possible images, adding a transformed copy $\mathbf{x}' = T_\phi \, \mathbf{x}$ of a given point $\mathbf{x}$ (*e.g.* rotating it with an angle $\alpha_\phi$ and adding small noise $\varepsilon_\phi$) means adding $(\mathbf{x}', l(\mathbf{x}))$ to the dataset, where $l(\mathbf{x})$ is the desired label for $\mathbf{x}$. But if $(\mathbf{x}', l(\mathbf{x}'))$ was already in the dataset, this amounts to enriching the possible labels for $\mathbf{x}'$. Supposing $T_\phi$ is an invertible transformation parameterized by $\phi$, full data augmentation (*i.e.* for all possible $\phi$, applied on all points $\mathbf{x}$) enriches $\mathbf{x}'$ with all labels $l(T_\phi^{-1}(\mathbf{x}'))$. In case of i.i.d. label noise, data augmentation will thus reduce this noise by a factor $\sqrt{\text{number of copies}}$.