[Reviews · NeurIPS 2019]

Reviewer 1



This main paper contribution is a similarity measure that at its core is the inner product of network gradients at two given data points, which distance we want to measure. It then develops the formula for shared-weights multiclass classifier so one can apply this measure to modern neural nets. The presented theory seems to be well established and original. The introduction, specifically the motivation part and similarity section, is quite clear and invites the reader to dive in into the details. Good job. The theory is developed step by step, from simple NNs to modern ones, in an easy to follow way. The paper might benefit from higher significance: The developed approach has a lot of potential insights; however, its primary purpose: to better understand the self-denoising effect, was weakly achieved. Moreover, experiments of successful applications of the method are lacking. One thing that worries me is that this statistic can be highly unstable during training and therefore not usable in practice. minor points: -Pg. 7 doesn't have all of its line numbers. -Double-Check math notations, as in pg. 7 var expression. -figure 4. doesn't seem to have a log scale. -section 2.4 notations are confusing, a network diagram/more explaining of notations or anything the authors find useful should help. -lines 124 -139 are unclear, specifically the rotation invariance and information metric aspect (and here also, missing line numbers). -More straightforward outline: after reading the introduction, one expects that after the theory of the measure and density estimation (sections 2-4), there will be an application section. The "enforcing similarity" (section 5) follows, which relates to the training stage and distracts from the main point. There are too many mini-topics, which makes it harder for the reader to follow: for example, in section 4, density estimation branches off to simulation description, overfit analysis, and uncertainty analysis, which I find confusing.

Reviewer 2



After reading the rebuttal, I tend to agree to changing my review as well conditioned on the following: 1: The structure of the paper should be revised to move the entire validation section to validation. The split validation section (indeed at least half of the validation is presented in the Introduction) is a large detriment to the quality of the paper. 2: An introduction section needs to be added to well frame the problem and mention related work. I don't like the "motivation-as-introduction" section as it does not present a wide scope within which the paper lies. 3: A clearer well presented argument as to precisely how the dataset self-denoising experiment is relevant. The rebuttal does a great job of this and it leaves me convinced of its strong application to real world datasets in general purpose training. Originality: The proposed method is highly novel and original. Within the scope of interpretability, there are no clear metrics to evaluate input similarity. This is further compounded as neural networks remain opaque which is increasingly becomes an issue as model interpretability becomes a leading concern. The related work is not well fleshed out and could use improvement. For example there may be a connection between the proposed method and influence functions. Consider also that generative models (in particular GANs as well) provide some notion of similarity in latent variable space. This connection deserves some highlighting. Quality: The submission is of mostly high quality. The proposed method has many useful theoretical properties that are well highlighted in the paper with insightful proofs and commentary. The empirical validation is a bit lacking as no clear use case is identified. Clarity: The paper is well written and well structured. It is easy to read and the content is well presented. The proofs are of high quality, and are easy to follow. Significance: The paper is of high significance. I think that the proposed approach and direction are both likely to yield further results. The proposed tool is likely to be useful as a visualization tool "as-is", however it is also likely to be useful as an interpretability tool in further work.

Reviewer 3



=== Post rebuttal === Thanks for addressing most of my concerns and adding experiments, and also great job at the added self-denoising analysis, I think it's a very cool application of the proposed measure! I'm updating my score from 4 to 6. === Original review === The paper is very clearly written and has a nice mixture of formal statements with intuitive explanations and examples. My main concern is with numerical experiments. There are 3 experiments: a) on a toy dataset to show that the similarity measure makes sense (and it does); b) on MNIST to show that one can use the proposed measure to speed up the training; c) on a dataset of satellite imagery to show that the proposed measure can be used to get insights about what's going on in the network and in the training process. However, the results on MNIST are not impressive at all (a tiny sped up on a very small dataset, and both methods converge to 80% validation accuracy, while on MNIST it should be around 98-99%). Also, none of the experiments has comparison against any baselines. At least the comparison against the perception losses (which is discussed in the paper to be similar but less justified, see lines 98-99) should be included. Experiment c) is supposed to show how to get insights by using the proposed measure, but I don’t understand what new knowledge/intuition about the network and the problem was obtained during the experiment. Similar to experiment a), experiment c) mainly just shows that the proposed measure makes sense, which is not enough to justify using it. Some less important points While theorem 1 is an interesting result and seem to motivate the (also very interesting) link to the perceptual loss (on line 98-99), it felt a bit left out: I’m not sure if this theorem was directly used anywhere throughout the paper. Not saying that it should be removed, but might be linked to other sections somehow? There might be interesting connections of the proposed measure to Deep image priors [1] (which, similar to the satellite images experiment, has the flavor of “denoising based on similarity between patches”) and to cycle-consistency papers for semi-supervised learning, e.g. [2, 3], which is close to the Group invariance idea mentioned on line 197. Calling the proposed measure “proper” compared to other measures seems a bit too far fetched. Line 191, there seem to be forgotten absolute value in the denominator [1] Ulyanov, Dmitry, Andrea Vedaldi, and Victor Lempitsky. "Deep image prior." Proceedings of the IEEE Conference on Computer Vision and Pattern Recognition. 2018. [2] P. Bachman, O. Alsharif, and D. Precup. Learning with pseudo-ensembles. In Advances in Neural Information Processing Systems, pages 3365–3373, 2014. [3] S. Laine and T. Aila. Temporal ensembling for semi-supervised learning. International Conference on Learning Representations, 2017.

[Author Response · NeurIPS 2019]

First, we would like to thank all reviewers for their very positive comments on the theory presented in the paper. This is a theoretical paper indeed, introducing a new concept, mathematically principled and studied. In order to be applicable in practice, we show how to compute it and how to quickly approximate it, with code available (on the anonymous github link provided in the paper). We also check experimentally that our estimator behaves correctly. We believe this is already a nice set of contributions.

In addition, we also propose a certain number of possible uses and extensions of this concept, showing it could be useful in many different ways. We consider it is out of the scope of this paper to actually run such applications, which would be difficult to include in the paper anyway for space reasons without sacrificing ideas in the theoretical section.

However, we do understand the main criticisms about: (a) the lack of insights brought by the large-scale experiment on remote sensing image registration in Section 6, and (b) the lack of comparison to the *perceptual loss*. For (b), we propose to add such a comparison on nearest neighbor retrieval in Section 6. We notice that the *perceptual loss* sometimes performs reasonably well, but often not. For instance, we show below the closest neighbors to a structured residential area image, for the *perceptual loss* (first row: does not make sense) and for our similarity measure (second row: similar areas).

To tackle (a), we propose to show how the similarity experimental computations in Section 6 can be used to **solve the initial problem**, by explicitly turning similarity statistics into a **quantification of the self-denoising effect**, as follows. Let us denote by $y_i$ the true (unknown) label for input $\mathbf{x}_i$, by $\widetilde{y}_i$ the noisy label given in the dataset, and by $\widehat{y}_i = f_\theta(\mathbf{x}_i)$ the label predicted by the network. We will denote the (unknown) noise by $\varepsilon_i = \widetilde{y}_i - y_i$ and assume it is centered and i.i.d., with finite variance $\sigma_\varepsilon$. The training criterion is $E(\theta) = \sum_j ||\widehat{y}_j - \widetilde{y}_j||^2$. At convergence, the training leads to a local optimum of the energy landscape: $\nabla_\theta E = 0$, that is, $\sum_j (\widehat{y}_j - \widetilde{y}_j)\nabla_\theta \widehat{y}_j = 0$. Let's choose any sample $i$ and multiply by $\nabla_\theta \widehat{y}_i$: using $k_\theta^I(\mathbf{x}_i, \mathbf{x}_j) = \nabla_\theta \widehat{y}_i . \nabla_\theta \widehat{y}_j$, we get: $\sum_j (\widehat{y}_j - \widetilde{y}_j) k_\theta^I(\mathbf{x}_j, \mathbf{x}_i) = 0$.

Let us denote by $k_\theta^{IN}(\mathbf{x}_j, \mathbf{x}_i) = k_\theta^I(\mathbf{x}_j, \mathbf{x}_i)\big(\sum_j k_\theta^I(\mathbf{x}_j, \mathbf{x}_i)\big)^{-1}$ the normalized kernel, and by $\mathbb{E}_k[a] = \sum_j a_j k_\theta^{IN}(\mathbf{x}_j, \mathbf{x}_i)$ the mean of value $a$ in the neighborhood of $i$, that is, the weighted average of the $a_j$ with weights $k_\theta^I(\mathbf{x}_j, \mathbf{x}_i)$ normalized to sum up to 1. This is actually a Parzen window estimator. Then the previous property can be rewritten as $\mathbb{E}_k[\widehat{y}] = \mathbb{E}_k[\widetilde{y}]$. As $\mathbb{E}_k[\widetilde{y}] = \mathbb{E}_k[y] + \mathbb{E}_k[\varepsilon]$, this yields: $\widehat{y}_i - \mathbb{E}_k[y] = \mathbb{E}_k[\varepsilon] + (\widehat{y}_i - \mathbb{E}_k[\widehat{y}])$

*i.e.* the difference between the predicted $\widehat{y}_i$ and the average of the **true labels** in the neighborhood of $i$ is equal to the average of the noise in the neighborhood of $i$, up to the deviation of the prediction $\widehat{y}_i$ from the average prediction in its neighborhood. **We want to bound the error** $\|\widehat{y}_i - \mathbb{E}_k[y]\|$ **without knowing** neither **the true labels** $y$ nor the noise $\varepsilon$. One can show that $\mathbb{E}_k[\varepsilon] \propto \mathrm{var}_\varepsilon(\mathbb{E}_k[\varepsilon])^{1/2} = \sigma_\varepsilon \|k_\theta^{IN}\|_{L2}$. The **denoising factor** is thus $\|k_\theta^{IN}\|_{L2}$, which is between $1/\sqrt{N}$ and 1, depending on the neighborhood quality. It is $1/\sqrt{N}$ when all $N$ data points are identical, i.e. all satisfying $k_\theta^C(\mathbf{x}_i, \mathbf{x}_j) = 1$. On the other extreme, this factor is 1 when all points are independent: $k_\theta^I(\mathbf{x}_i, \mathbf{x}_j) = 0 \ \ \forall i \neq j$. This way we extend *noise2noise*[11] to real datasets with non-identical inputs. **In our remote sensing experiment**, we estimate this way a denoising factor of 0.02, consistent across all training rounds and inputs ($\pm 10\%$), implying that each training round contributed equally to denoising the labels. This is confirmed by Fig. 2, which shows the error steadily decreasing, on a control test where true labels are known. The shift $(\widehat{y}_i - \mathbb{E}_k[\widehat{y}])$ on the other hand can be directly estimated given the network prediction. In our case, it is 4.4px on average, which is close to the observed median error for the last round in Fig. 2. It is largely input-dependent, with variance 3.2px, which is reflected by the spread distribution of errors in Fig. 2. This input-dependent shift thus provides a hint about prediction reliability.

It is also possible to bound $(\widehat{y}_i - \mathbb{E}_k[\widehat{y}]) = \mathbb{E}_k[\widehat{y}_i - \widehat{y}]$ using only similarity information (without predictions $\widehat{y}$). **Theorem 1** implies that the application: $\frac{\nabla_\theta f_\theta(\mathbf{x})}{\|\nabla_\theta f_\theta(\mathbf{x})\|} \mapsto f_\theta(\mathbf{x})$ is well-defined, and it can actually be shown to be differentiable and Lipschitz with a network-dependent constant. Thus $\|f_\theta(\mathbf{x}) - f_\theta(\mathbf{x}')\| \leqslant C \left\| \frac{\nabla_\theta f_\theta(\mathbf{x})}{\|\nabla_\theta f_\theta(\mathbf{x})\|} - \frac{\nabla_\theta f_\theta(\mathbf{x}')}{\|\nabla_\theta f_\theta(\mathbf{x}')\|} \right\| = \sqrt{2}C\sqrt{1 - k_\theta^C(\mathbf{x}, \mathbf{x}')}$, yielding $\|\widehat{y}_i - \widehat{y}_j\| \leqslant \sqrt{2}C\sqrt{1 - k_\theta^C(\mathbf{x}_i, \mathbf{x}_j)}$ and thus $\mathbb{E}_k[\widehat{y}_i - \widehat{y}] \leqslant \sqrt{2}C \, \mathbb{E}_k\left[\sqrt{1 - k_\theta^C(\mathbf{x}_i, \cdot)}\right]$.

**Other:** thank you for the very relevant literature, and the nice application suggestion to GAN / cycle-consistency. We can postpone the paragraph *Dynamics of learning* to the appendix to make place for the section above if needed.

[Meta-Review · NeurIPS 2019]

All of the reviewers found the proposed technique original and the theory interesting. The reviewers initially had concerns regarding the structure of the paper, relevance of some of the experiments, and comparison with perceptual loss. These concerns are alleviated given the author feedback. Assuming that the authors will integrate the author feedback into the paper and incorporate all of reviewers' feedback, I recommend acceptance as a poster.